



# Spatial distribution and variability of boundary layer aerosol particles observed in Ny-Ålesund during late spring in 2018

Barbara Harm-Altstädter[1], Konrad Bärfuss[1], Lutz Bretschneider[1], Martin Schön[2], Jens Bange[2],
Ralf Käthner[3], Radovan Krejci[4], Mauro Mazzola[5], Kihong Park[6], Falk Pätzold[1], Alexander Peuker[1],
Rita Traversi[5,7], Birgit Wehner[3], and Astrid Lampert[1]

[1]Institute of Flight Guidance, Technische Universität Braunschweig, 38108 Braunschweig, Germany
[2]Center for Applied Geosciences, Eberhard Karls University Tübingen, 72076 Tübingen, Germany
[3]Department of Experimental Aerosol and Cloud Microphysics, Leibniz Institute for Tropospheric Research, 04318 Leipzig, Germany
[4]Department of Environmental Science, Stockholm University, 10691 Stockholm, Sweden
[5]Institute of Polar Sciences, National Research Council, 40129 Bologna, Italy
[6]School of Earth Sciences and Environmental Engineering, Gwangju Institute of Science and Technology, 61005 Gwangju, Republic of Korea
[7]Department of Chemistry "Ugo Schiff", University of Florence, 50019 Sesto F.no, Italy

**Correspondence:** Barbara Harm-Altstädter (b.altstaedter@tu-braunschweig.de)

**Abstract.** This article aims to improve the understanding of the small scale aerosol distribution affected by different atmospheric boundary layer (ABL) properties. In particular, transport and mixing of ultrafine aerosol particles (UFP) are investigated, as an indicator for possible sources triggering the appearance of new particle formation (NPF) at an Arctic coastal site. For this purpose, flexible measurements of unmanned aerial systems (UAS) are combined with continuous ground based obser-
vations at different altitudes, the observatory Gruvebadet close to the fjord at an altitude of 67 m above sea level (a.s.l.), and the observatory at the Zeppelin Mountain at an altitude of 472 m a.s.l.. The two unmanned research aircraft called ALADINA and MASC-3 were applied for field activities at the polar research site Ny-Ålesund, Svalbard, between 24 April 2018 and 25 May 2018. The period was at the end of Arctic haze during the snow melt season. A high frequency of occurrence of NPF was observed, namely on 55 % of the airborne measurement days. With ALADINA, 230 vertical profiles were performed between the
surface and the main typical maximum height of 850 m a.s.l., and the profiles are connected to surface measurements, in order to obtain a 4-D picture of aerosol particle distribution. Analyses of potential temperature, water vapour mixing ratio and aerosol particle number concentration of UFP in the size range of 3–12 nm ($N_{3-12}$) indicate a clear impact of the ABL's stability on the vertical mixing of the measured UFP, which results in systematical differences of particle number concentrations at the two observatories. In general, higher concentrations of UFP occurred near the surface, suggesting the open sea as the main source
for NPF. Three different case studies show that the UFP were rapidly mixed in the vertical and horizontal scale depending on atmospheric properties. In case of temperature inversions, the aerosol population stayed confined to specific altitude ranges, and was not always detected at the observatories. However, during another case study that was in relation to a persistent NPF event with subsequent growth rate, the occurrence of UFP was identified to be a wide spreading phenomenon in the vertical scale, as the observed UFP exceeded the height of 850 m a.s.l.. During a day with increased local pollution enhanced equivalent



black carbon mass concentration (eBC) coincided with an increase of the measured $N_{3-12}$ in the lowermost 400 m, but without subsequent growth rate. The local pollution was transported to higher altitudes, as measured by the UAS. Thus, emissions from local pollution may play a role for potential sources for UFP in the Arctic as well. In summary, a highly variable spatial and temporal aerosol distribution was observed with small scales at the polar site Ny-Ålesund, determined by atmospheric stability, contrasting surface and sources, and topographic flow effects. The UAS provides the link to understand differences measured

at the two observatories at close distance, but different altitudes.

## 1 Introduction

The interactions between formation, growth, transport, and vertical mixing of aerosol particles in the atmosphere need a more profound understanding, especially in the Arctic atmospheric boundary layer (ABL). In general, the Arctic is affected by a warming rate of the surface air temperature twice as high in comparison with the global average (IPPC, 2013), an effect

well known as "Arctic amplification" (AA, Serreze and Barry, 2011). The phenomenon implies vast changes in the feedback processes between the atmosphere and cryosphere (sea ice, snow, ice), mostly affected by and resulting in a rapid decrease of the sea ice extent and sea ice thickness (e.g., Stroeve et al., 2012; Dai et al., 2019). However, future scenarios of the Arctic climate are still not clear (e.g., Screen et al., 2018) and more observations are essential in order to better characterize the feedback mechanisms of the AA (Wendisch et al., 2017, 2022). Besides the main contributors of surface albedo, mixed phase

clouds and sea ice extent (e.g., Vavrus, 2004; Taylor et al., 2013; Zhang et al., 2018), aerosol particles are considered to play a key role in the AA (Serreze and Barry, 2011). This means, for instance, a direct effect of the aerosol particles on the Earth's radiation budget (Twomey, 1991; Haywood and Boucher, 2000), which is mainly triggered by the number concentration and chemical composition of the particles. In this context, carbonaceous aerosol particles like black carbon (BC) are of particular relevance, as BC strongly absorbs in the visible spectrum of the solar radiation, which ultimately leads to an increase of the

ambient temperature (e.g., Bond et al., 2013). Additionally, the snow albedo might be reduced after deposition of BC on the snow covered or frozen surfaces (Flanner et al., 2009), and aged carbonaceous particles may also have the potential to enhance cloud cover, as they can act as cloud condensation nucleis (CCN) or ice nucleis (IN). This might further reinforce the AA, as low level clouds tend to warm the Arctic surface (Zhao and Garrett, 2015), except for short periods in the summer months (e.g., Intrieri et al., 2002; Kay and L'Ecuyer, 2013). However, the significance and magnitude of feedback mechanisms, initiated by

the presence of aerosol particles in the Arctic, are still subject to current debates (e.g., Pithan and Mauritsen, 2014; He et al., 2019; Schmale et al., 2021).

This is also a reason why a deeper knowledge of the role of new particle formation is of crucial importance in the Arctic, as by subsequent growth, ultrafine aerosol particles (UFP or nanoparticles, size < 50 nm) can modify directly the radiation budget or act as CCN (Kerminen et al., 2012) as well, and may therefore indirectly impact the Earth's radiation budget. Although

median growth rates of 2.3 nm h$^{-1}$ are low at Arctic research sites and comparable to boreal forest observations (Kerminen et al., 2018), NPF was frequently observed during the summer season with maximum aerosol particle concentrations of several 1000 cm$^{-3}$ (e.g., Ström et al., 2009; Tunved et al., 2013; Freud et al., 2017). Currently, a large diversity of different factors



contributing to new particle formation in the Arctic environment are known, where the most important factor, the intensity of the solar radiation (Kerminen et al., 2018; Nieminen et al., 2018), is of minor relevance in comparison to mid latitudes due to

the lower solar elevation angles in the polar regions. It is still under discussion if UFP generally originate from new particle formation after subsequent growth of the gas-particle phase, like it was found by Wiedensohler et al. (1996) in the Arctic maritime ABL during summer and autumn. Tunved et al. (2013) presumed that new particle formation is likely formed locally due to photo-chemical production, as increased number concentrations of UFP were observed during the summer months with the highest incoming solar radiation in Spitsbergen (Norway). But the authors of the last mentioned study take into account

another possibility and hypothesize that the observed UFP may have been entrained from the free troposphere (FT) and were possibly transported to the measurement site, as the measurements were carried out at the Zeppelin Observatory at a height of 472 m above sea level (a.s.l.) in Ny-Ålesund. The observations lead to the assumption that UFP may have originated from aloft, most likely caused by high turbulence in the entrainment zone (EZ) that can trigger new particle formation, as it was recognized previously for instance by Nilsson et al. (2001). Heintzenberg et al. (2017) analyzed a 10-year data set of new

particle formation in the Svalbard area and excluded a potential connection to Arctic haze, and presumed marine biological activity as a source for precursor gases of new particle formation due to photo-chemical reactions in summer time. However, new particle formation was observed earlier in spring time as well and Dall'Osto et al. (2017) hypothesized sea ice melt as possible trigger, as a clear connection was found between nucleation days and highest ammonia gas concentrations ($NH_3$). The idea is that biological precursor gases are emitted from the Arctic Ocean after sea ice melt during spring. But so far, the

sources for atmospheric ammonia are still unclear and some studies suggested sea bird colonies as a possible emission hot spot that might lead to the ternary nucleation process for new particle formation (Blackall et al., 2007; Riddick et al., 2012; Croft et al., 2016). Other studies consider iodines as a major source for new particle formation at high latitude coastal areas (e.g., Allan et al., 2015; Sipilä et al., 2016), as well as iodic acid ($HIO_3$) that was observed with highest rates above pack ice in the central Arctic Ocean (Baccarini et al., 2020). In addition to this, dimethyl sulfide (DMS) is supposed to play an important role

for new particle formation after its oxidation to methane sulfonic acid (MSA) and sulfuric acid ($H_2SO_4$) (Leaitch et al., 2013). For present scenarios in the maritime ABL, the availability of DMS may be one of the most dominant factors as precursor gas for primary sulfate aerosol particles in the Arctic, especially related to the ongoing rapid decrease of the sea ice extent which might further accelerate the release of DMS (e.g., Gabric et al., 2005). This was recently verified by Lee et al. (2020) who further support the assumption of a local origin for new particle formation in Ny-Ålesund.

However, it is difficult to accurately determine the local source of UFP in the Arctic ABL due to the limited number of measurements available for small particle sizes, in particular for 1 to 2 nm. Additionally, new particle formation in the ABL may be influenced by a combination of various factors that occur simultaneously on different scales. The lack of knowledge about the nucleation process and subsequent growth of aerosol particles in the vertical and horizontal distribution contributes to the uncertainty surrounding the role of aerosol particles on the AA. This is largely due to limited data availability in the Arctic

region, caused by high cost and difficulty of access to research sites. There are data gaps in consequence of the limited availability of suitable measurement methods that would allow for frequent profiling between the surface and the FT in the ABL. A comprehensive understanding of the life cycle of aerosol particles is crucial in identifying potential sources of new particle for-





mation. To accomplish this, it is important to conduct spatio temporal UFP measurements, encompassing nucleation, growth, and mixing within the ABL. At this point, the use of suitably equipped unmanned aerial systems (UAS) has a high potential for achieving a better understanding of the spatial distribution of aerosol particles in relation to different ABL properties. The large flexibility is one of the main advantages of the UAS compared to tethered balloons or radiosondes that were used for vertical profiling during several studies in Ny-Ålesund (e.g., Moroni et al., 2015; Ferrero et al., 2016). The recently published report of Hann et al. (2021) summarized UAS applications that were previously carried out in Svalbard and provided a detailed overview of rapidly growing applications within the last few years, but activities in atmospheric research played a minor role. In particular the spatial distribution of UFP was not studied so far, but would be essential in order to document possible sources for new particle formation. Processes above land in comparison with processes above open water or sea ice can be investigated by UAS operations as low altitudes. In this article, the focus is on the results of aerosol observations rather than on the technical background of the campaign that was already introduced in Lampert et al. (2020), and a general review of UAS campaigns in Svalbard is not in the scope of this publication and other case studies of the ALADINA period have already been subject to publications shown in Lampert et al. (2020), Petäjä et al. (2020), Schön et al. (2022a) and Xavier et al. (2022).

The aim of this article is to present an overview of the UAS field campaign and the gained data, in order to better understand the horizontal and vertical variability of aerosol particles in relation to the Arctic ABL. One of the main advantages of the UAS is to link observations between different research sites, here the Zeppelin Observatory (ZEP, 78°56' N, 11°53' E, 472 m a.s.l.) and the Gruvebadet facility (GRU, 78°55' N, 11°56' E, 67 m a.s.l.) that provide long term aerosol measurements at different altitudes. A connection between both stations is of vital importance to characterize dynamic effects like vertical mixing and horizontal transport on small scales, and to assess the role of the ABL's stability on the spatial distribution of UFP.

The article is structured as follows: Sect. 2 provides an overview of the research area around Ny-Ålesund, the aerosol instrumentation at Gruvebadet and Zeppelin Observatory as well as onboard the UAS, the methods and data availability during the experiment. The results are presented in Sect. 3, starting with a campaign overview of aerosol observations at the two research sites in comparison with vertical profiles of UFP with a diameter in the size range of 3 and 12 nm ($N_{3-12}$) derived from ALADINA. In a summary, 230 vertical profiles of the aerosol particle number concentrations for different sizes and meteorological parameters like potential temperature $\theta$ and water vapour mixing ratio $r$ are discussed in order to assess a correlation between the occurrence of UFP and ABL properties. In addition, three selected case studies are presented in more detail that focus on different aspects. The case studies comprise observations during the end of the Arctic haze period from 24–26 April 2018 (Case I), high variability of UFP in the horizontal scale during a nucleation event on 20 May 2018 (Case II) and a study of eBC measured during a day affected by a higher degree of local pollution on 23 May 2018 (Case III). This study ends with a conclusion in Sect. 4.



## 2 Description of the measurement site, instrumentation and data availability

### 2.1 Research site Ny-Ålesund

The topography around the international research area of Ny-Ålesund combines a highly variable terrain with tundra, hills, mountains, glaciers, fjords and the Arctic Ocean on small distances of a few 100 m (see Fig. 1). The village of Ny-Ålesund (78°55' N, 11°52' E, 11 m a.s.l.; Fig. 1) belongs to the Svalbard Islands and is located at the southern coast of the Kongsfjord, south westerly at a distance of around 10 km away from the Kongsvegen glacier. The fjord is orientated in the north-west/south-east axis and defines the two main wind regimes at the measurement area in Ny-Ålesund. One wind sector originates from the

flow from the direction of the Kongsvegen glacier, leading to high wind speed from east to south-east. The other wind regime is from south to south-west from the Arctic Ocean.

During summer, there is also a frequent north-west to south-west wind, caused by drainage flows from the Zeppelin Mountain to the fjord and low wind speed from the open sea (e.g., Beine et al., 2001; Mazzola et al., 2016). However, the wind regimes are mainly valid for the lowermost 500 m (Graßl et al., 2022), like measured at Old Pier or at the Gruvebadet observatory that

is also influenced by a katabatic flow from the Broggerbreen in the west (Schön et al., 2022a). The latter is situated south-west of the village of Ny-Ålesund and south-east of the airfield at a respective distance of around 1 km (see Fig. 1). Beine et al. (2001) showed that wind speed and wind direction are different at Zeppelin Observatory, which is located on the top of Mount Zeppelin at a distance of around 2.3 km south of the village. During most time, the station is within the ABL, but to a minor part observations represent conditions of the lowermost FT (e.g., Tunved et al., 2013). During spring, the research

station is mainly influenced by southerly wind, so that possible local pollution from the village should be of minor importance during the general highest research activity in Ny-Ålesund (Beine et al., 1996). This was recently verified by Dekhtyareva et al. (2018) who further investigated a significant non-linearity of the measured temperature between the Zeppelin Observatory and observations close to sea level altitude, in consequence of the different altitude levels and complex terrain. The effect was particularly observed during the summer months, most likely caused by wind shear, as a result of different air flows that

typically occur within the lowermost 500 m, and above 800 m the wind direction tends to merge into the synoptic flow (Graßl et al., 2022). Apart from a high impact of the topography on meteorological properties, the site is characterized by a high variability in the aerosol composition as well. For instance, Ström et al. (2003) and Tunved et al. (2013) showed a seasonal variability of the aerosol particle mode measured at the Zeppelin Observatory. In principle, the spring months (March–May) are dominated by accumulation mode particles, that mainly originate from long range transport outside of the Arctic, a phenomenon

called "Arctic haze". The summer months (June–August) show a minor role of accumulation mode particles and a domination of the nucleation mode, mainly linked to a low condensation sink (CS), referring to Park et al. (2017). During the rest of the year (September–February), the site is influenced by a low number concentration of accumulation mode particles and also by a minor relevance of nucleation mode particles with an overall minimum in September/October.





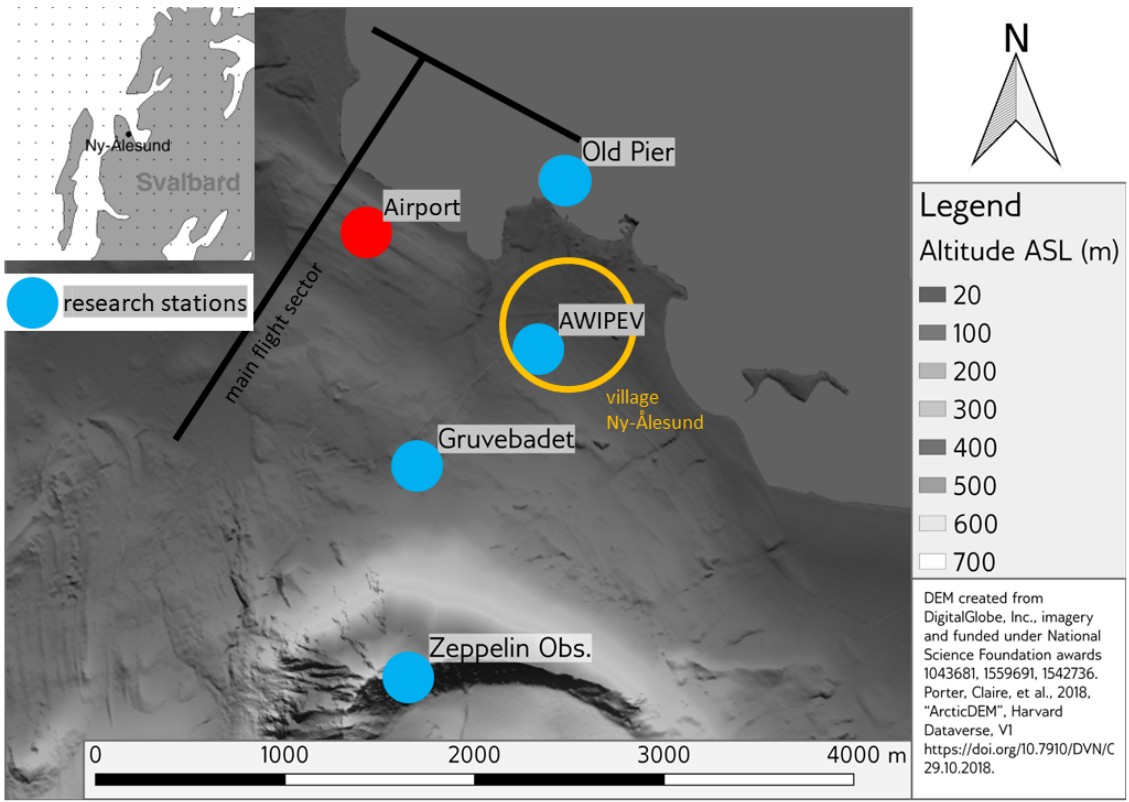

**Figure 1.** The map represents the topography (grey shading in colour bar) of the investigation area around Ny-Ålesund that belongs to the Svalbard Islands, Norway. Research flights were performed with the two UAS ALADINA and MASC-3 at the local airport (red point, appr. 40 m a.s.l.) in April–May 2018. The main flight patterns were performed in parallel to the airport and crossing the coast via horizontal legs (black lines). Aerosol in-situ data are used from Gruvebadet (67 m a.s.l.) and Zeppelin Observatory (472 m a.s.l.) and meteorological data is taken into account from the AWIPEV station, within the research village that is indicated by the yellow circle. Future planned studies will consider turbulent flux measurements at Old Pier as well. The main wind directions are represented by the white arrows: one from the Kongsfjord glacier (SE) and a temporary wind direction coming from the glaciers, south of the village (SSW).

## 2.2 Aerosol monitoring

In situ observations of aerosol particles are taken into account from the Gruvebadet research station (GRU) and the Zeppelin Observatory (ZEP), see Fig. 2a. GRU data represent surface measurements, and observations at ZEP are mainly representative for conditions at the higher parts of the ABL, and to a minor fraction in a transition zone between the ABL and FT.

Further, UFP with a size of 3-12 nm are determined with a nano-SMPS (nano-scanning mobility particle sizer) deployed at ZEP which is a combination of a nano-DMA (differential mobility analyzer, model 3085, TSI Inc., USA) and a CPC
(condensation particle counter, model 3776, TSI Inc., USA) in 3 min temporal intervals.





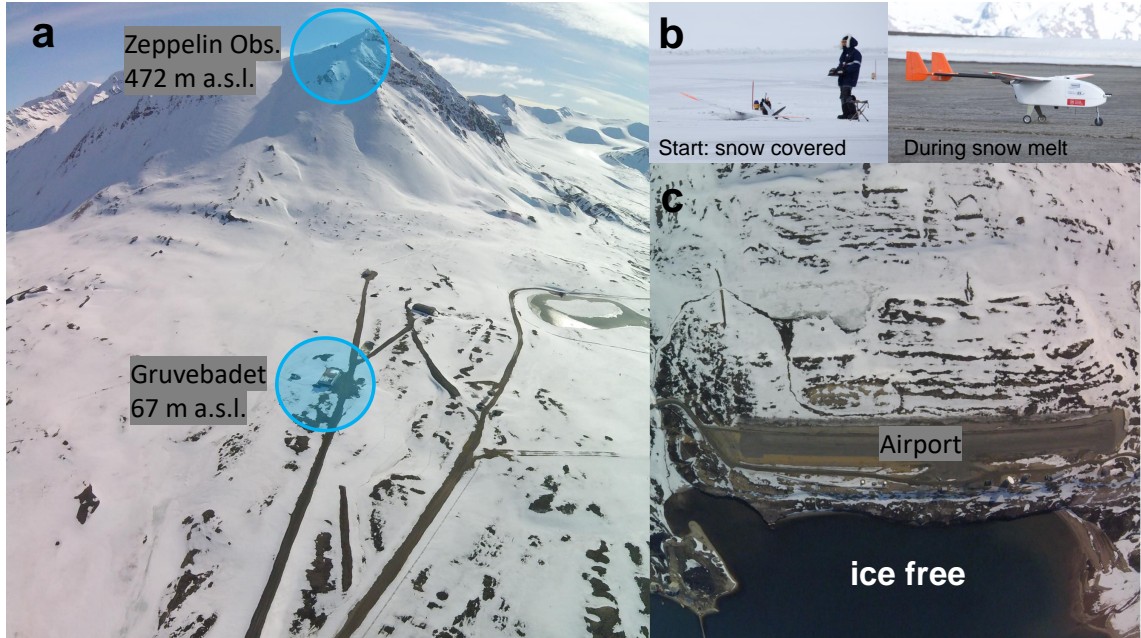

**Figure 2.** For a better orientation and information on snow conditions, selected pictures are shown of the investigation area which were taken during the field period. (a) The two research stations Gruvebadet and Zeppelin Observatory from a bird's eye view, taken at the end of the campaign on 25 May 2018. The UAS MASC-3 was used for studying turbulent properties (b, left hand-side) and ALADINA was operated for investigating of aerosol particles linked to atmospheric boundary layer properties (b, right hand-side). At the beginning of the campaign, the site was covered with snow but the snow melt occurred and the water area around the coast was completely ice free (c). Pictures: ©TU Braunschweig

In order to provide information about possible local pollution at the investigation site eBC mass concentration data are used, which are calculated from the aerosol light absorption coefficient measured with a multi-angle absorption photometer (MAAP, model 5012, Thermo Fisher Scientific Inc., USA). The Zeppelin Observatory in its full facility was recently presented in Platt et al. (2022) and shows more information about the instrumentation available at site.

A scanning mobility particle sizer (SMPS, model 3034, TSI Inc., USA) is deployed at GRU, which measures in the particle size distribution between 10 and 470 nm (Hogrefe et al., 2006; Lupi et al., 2016). At ZEP, the aerosol size distribution is derived from a combination of differential mobility particle sizers (DMPS) in the size of 5–810 nm and 10–790 nm.

### 2.3 Unmanned aerial systems (UAS)

Research flights with two UAS were performed at the local airfield in Ny-Ålesund (see Fig. 1 and Fig. 2c). In general, the

measurement flights were orientated in parallel to the airport and perpendicular over open water areas near the coast (see Fig. 1) in order to investigate the horizontal distribution of aerosol particles and meteorological parameters above different surface conditions. Both UAS are fixed wing aircraft developed for atmospheric research with a take-off weight smaller than 25 kg



and electrically powered. The cruising speed is less than $30\,\mathrm{m\,s^{-1}}$, which further results in a high temporal resolution of the measured data in comparison with a typically faster cruising speed of manned aircraft with around $60–70\,\mathrm{m\,s^{-1}}$. In addition, the

two UAS are equipped with autopilot systems and are automatically controlled during measurement flights after programming a well-defined flight path of the flight missions. One major challenge of the UAS application was the restricted frequency use of $> 2\,\mathrm{GHz}$, as Ny-Ålesund is a radio silent zone. For this purpose, all modules had to be adapted from the typical $2.4\,\mathrm{GHz}$ and were run to work at 433 and $868\,\mathrm{MHz}$, respectively. In the following, both systems are briefly introduced, as they differ in their designs and payloads.

### 175 2.3.1 UAS ALADINA

The UAS ALADINA (Application of Light-weight Aircraft for Detecting IN-situ Aerosol, Fig. 2b) is based on the aircraft family of type Carolo P360 and was designed at the Technische Universität Braunschweig. The fixed wing airplane has a wing span of $3.6\,\mathrm{m}$, a take-off weight of $24.8\,\mathrm{kg}$ and a mean flight duration of $35–45\,\mathrm{min}$. Its first performance was described in Altstädter et al. (2015), but for the polar application shown here, the design of ALADINA and its instrumentation on-board have

undergone fundamental changes, which are presented in Lampert et al. (2020). The aircraft is insulated, and batteries are stored in a side board, which guarantees a fast turnaround time of around $20\,\mathrm{min}$, as batteries are easily replaceable in the field and data can be quickly saved via downlink after each landing. The inner compartment is heated to a stable temperature range in order to assure the reliability of the instruments' working flow, which is especially important for the aerosol sensors. The payload weighs around $4.5\,\mathrm{kg}$ and consists of meteorological sensors, aerosol instrumentation and batteries for measurement devices.

Different types of temperature sensors and humidity sensors, as well as a multi-hole probe are installed for the calculation of air temperature, humidity and the 3-D wind vector, and the sensors are mounted at the tip of the aircraft in order to assure undisturbed measurements of the air probe. In addition, two pyranometers are integrated into the UAS, one on top and the other one underneath on the fuselage, which enable measurements of the incoming solar radiation and reflex radiation, respectively. A detailed description of the meteorological measurement unit is not in the scope of the current study, as information are

provided in Bärfuss et al. (2018) and Lampert et al. (2020). The performance of the newly designed aerosol instrumentation environment is also shown in Lampert et al. (2020).

Two condensation particle counters of the same type (CPC, model 3007, TSI Inc., USA) are used with different threshold diameters, which allow measurements of the aerosol particle number concentrations with a size of up to around $1\,\mu\mathrm{m}$. The two CPCs were tested and modified by TROPOS (Leibniz Institute for Tropospheric Research) and are tuned down to cut

off sizes of $3\,\mathrm{nm}$ (CPC1) and $12\,\mathrm{nm}$ (CPC2), respectively. Thus, the study presented here takes into account the observed aerosol particle number concentration of UFP in the size range between 3 and $12\,\mathrm{nm}$, hereafter referred to $N_{3-12}$, within an uncertainty of $\pm20\,\%$ at $1\,\mathrm{s}$ temporal resolution (Altstädter et al., 2015). An optical particle counter (OPC, model GT-526S, Met One Instruments Inc., USA) measures the larger particles in six size channels. In this article, only one out of the total six size channels is considered, valid for particles with a size between 300 and $500\,\mathrm{nm}$ ($N_{300-500}$), as larger particles were not

detectable during the investigation period. The concentrations have a measurement error of $\pm\,15\,\%$ (Altstädter et al., 2015). The flow system of the original handheld instruments has been modified by substituting the internal pumps with a single, more



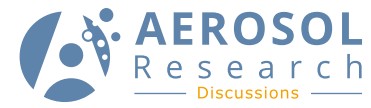

powerful one (diaphragm pump 1420VP BLDC, Gardner Denver Thomas GmbH, Germany) and implementing orifices after the detectors, which are driven critically. The orifice diameters for the two CPCs are $5.1 \times 10^{-3}$ inches and provide a volume flow of approximately $125 \, \text{ml min}^{-1}$ under standard atmospheric conditions and $2.0 \times 10^{-2}$ inches for the OPC optics, which

results in a volume flow of approximately $2 \, \text{l min}^{-1}$.

Additionally, a micro aethalometer (microAeth® model AE51, $\lambda$=880 nm, AethLabs, USA) is implemented onboard for detecting the equivalent black carbon (eBC) mass concentration, based on the light absorbing measurement principle. The data handling and post-processing of the calculated eBC is equivalent to the performance presented in Altstädter et al. (2020). The AE51 is susceptible to humidity and temperature gradients (Altstädter et al., 2020) and its reliability is limited by artefacts in

the attenuation signal that mainly correlate with a small aerosol background concentration (e.g., Pikridas et al., 2019) within a given accuracy of $\pm 10 \, \%$, as stated by the manufacturer. Regarding a previous field campaign with ALADINA in West-Africa, the uncertainty was calculated to $\pm 200 \, \text{ng eBC m}^{-3}$ for a temporal resolution of 1 Hz. This is a critical point for the measurement reliability of the AE51 in the Arctic, firstly as the background aerosol number concentration is low in Svalbard, with around several $100 \, \text{cm}^{-3}$ (Tunved et al., 2013), and secondly, the eBC load is marginal and far below the specified

detection limit of the AE51. For instance, a maximum of around $80 \, \text{ng m}^{-3}$ was measured in NyÅlesund between the years of 1998 and 2007 (Eleftheriadis et al., 2009), derived from an aethalomter of model AE31 which works at the same wavelength of $\lambda$=880 nm as the AE51. In consequence of an expected limited performance of the AE51, eBC measurements are not provided in a statistical analysis in the manuscript shown here.

### 2.3.2 UAS MASC-3

The UAS MASC-3 (Multi-Purpose Airborne Sensor Carrier) in its third version (Fig. 2b) was developed by the Eberhard Karls Universität Tübingen and is equipped with a sensor system for meteorological measurements and described in more detail in Rautenberg et al. (2019). MASC-3 has a wingspan of 4 m, a weight of 5–8 kg, depending on the payload, and a maximum flight duration of 2 h. The sensor system consists of a multi-hole probe, a fine-wire platinum resistance thermometer and a slower digital humidity sensor. The high resolution calculated 3-D wind vector and air temperature can resolve turbulent fluctuations.

For the field campaign in Ny-Ålesund, some adaptions had to be undergone. For instance, all heated electronic parts were insulated in the hull with foam to maintain a stable temperature. The batteries were pre-heated before take-off and insulated in foam in order to assure a warm temperature that is essential for safety reasons and for a long flight duration under cold ambient conditions.

### 2.4 UAS flights in Ny-Ålesund and data availability

Table 1 provides an overview of the individual measurement days and the data availability of the UAS ALADINA in Ny-Ålesund between 24 April 2018 and 25 May 2018. This includes information about the total number of vertical profiles that were performed with ALADINA for the specific measurement days, flight operation with MASC-3, as well as aerosol data measured with SMPS at GRU, nano-SMPS, DMPS and MAAP at ZEP. During the investigation period, 49 research flights were operated with ALADINA on eleven different measurement days (see Tab. 1, Fig. 3b and Fig. 4b) which led to a sampling



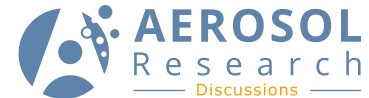

**Table 1.** This table summarizes the data availability of the study presented here. The UAS ALADINA performed 49 research flights on eleven different days, and a total sample of 230 vertical profiles was enabled during the polar field campaign in Ny-Ålesund between 24 April and 25 May 2018. The second UAS MASC-3 was operated during six common measurement days with ALADINA, and the study considers vertical profiles of the calculated horizontal wind. Ground based aerosol data is continuously available from SMPS at the Gruvebadet Observatory (GRU). Aerosol data measured with DMPS and nano-SMPS are not complete on a daily basis at the Zeppelin Observatory (ZEP) during the period. Equivalent black carbon mass concentrations (eBC) is calculated from MAAP at the Zeppelin Observatory. "NO" means not operated, "NA" stands for not available, "X" represents data availability during the individual ALADINA measurement flights and the data are freely accessible in Harm-Altstädter et al. (2022).

| Measurement day | UAS | | GRU | ZEP | | |
| | Profiles[a] | MASC-3 | SMPS | nano-SMP | DMPS | MAAP |
| --- | --- | --- | --- | --- | --- | --- |
| 24-04-2018 | 4 | X | X | X | X | X |
| 25-04-2018 | 6 | NO | X | X | X | X |
| 26-04-2018 | 6 | X | X | X | X | X |
| 01-05-2018 | 24 | X | X | X | X | X |
| 14-05-2018 | 10 | NO | X | X | X | X |
| 15-05-2018 | 40 | NO | X | X | X | X |
| 19-05-2018 | 32 | NO | X | NA | X | X |
| 20-05-2018 | 23 | NO | X | NA | NA | X |
| 21-05-2018 | 31 | X | X | NA | NA | X |
| 23-05-2018 | 21 | X | X | NA | X | X |
| 25-05-2018 | 33 | X | X | NA | NA | X |

[a] Number of vertical profiles that were performed with the unmanned aerial system (UAS) ALADINA within the indicated measurement day. In total, 230 vertical profiles were enabled during the measurement period and a summary of all profiles is subject to the analysis shown in Figs. 7–8 and for a better orientation of the situation the profiles are presented in Figs. A1–A6 according to the time series of the individual parameters.

time of around 29 h. In total, 230 vertical profiles and around 300 horizontal transects (mainly at the heights of 150, 300 and 450 m a.s.l.), called legs, were carried out during the field experiment.

Horizontal flights were mainly performed with ALADINA in the last week of the campaign between 18–25 May 2018, and as the priority of the flight mission was on vertical profiling with a typical maximum altitude of 850 m a.s.l., the horizontal legs are on short distance in order to enable as many vertical profiles as possible during one measurement flight limited by the

batteries capacity. The mean flight duration was around 35 min. For this reason, turbulent properties are not considered with ALADINA, as they require multiple horizontal flight patterns at constant altitude for guaranteeing statistical relevance.

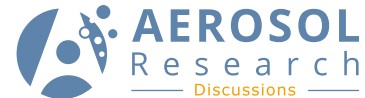

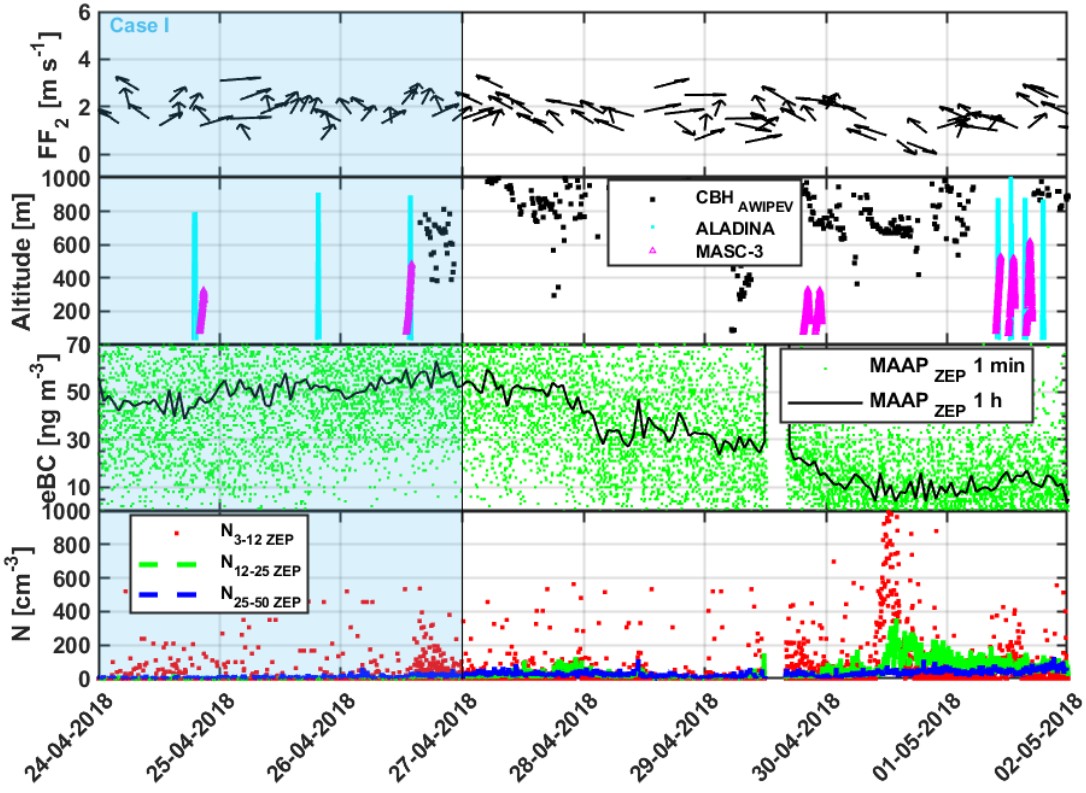

**Figure 3.** Time series of selected parameters that contribute to the UAS study in Ny-Ålesund, valid for the period between 00:00 UTC on 24 April and 00:00 UTC on 2 May 2018. From top to bottom: wind speed $FF_2$ in $\mathrm{m\,s^{-1}}$ calculated in a 2 h average, measured at the AWIPEV station at 2 m level (Maturilli, 2018a, b). The wind arrows represent the wind direction in the same time range, respectively. Cloud base height (CBH) in m for 10 min interval (Maturilli, 2018c, d) derived from the AWIPEV station (black dot) in comparison with the sequences of ALADINA profiles (cyan dot) and levels of horizontal legs based on MASC-3 (magenta triangle). Equivalent black carbon mass concentration (eBC) is estimated from a MAAP (Multi-Aerosol Absorption Photometer) at the Zeppelin Observatory in 1 min (green dot) and averaged for 1 h (black line). Aerosol particle number concentration (N) in the size of 3 to 12 nm ($N_{3-12}$, red dot), 12-25 nm ($N_{12-25}$, green dashed line) and 25-50 nm ($N_{25-50}$, blue dashed line) measured with a nano-SMPS in 3 min intervals at the Zeppelin Observatory. The blue shading represents the three measurement days of ALADINA that are considered for a deeper analyses in the first case study (Case I, Sect. 3.3).

However, this flight procedure was realized with the second UAS MASC-3 that was operated to a large degree at the same time on six common measurement days (see Tab. 1, Fig. 3b and Fig. 4b). A total of 13 quality assured research flights was performed on seven different measurement days with a sampling time of around 17 h. The MASC-3 flight periods are summarized in the study of Schön et al. (2022a). A typical measurement flight consists of horizontal legs with a length of at least 1.5 km. The legs are repeated three to four times at each measurement altitude, typically between 50 and 600 m a.s.l.

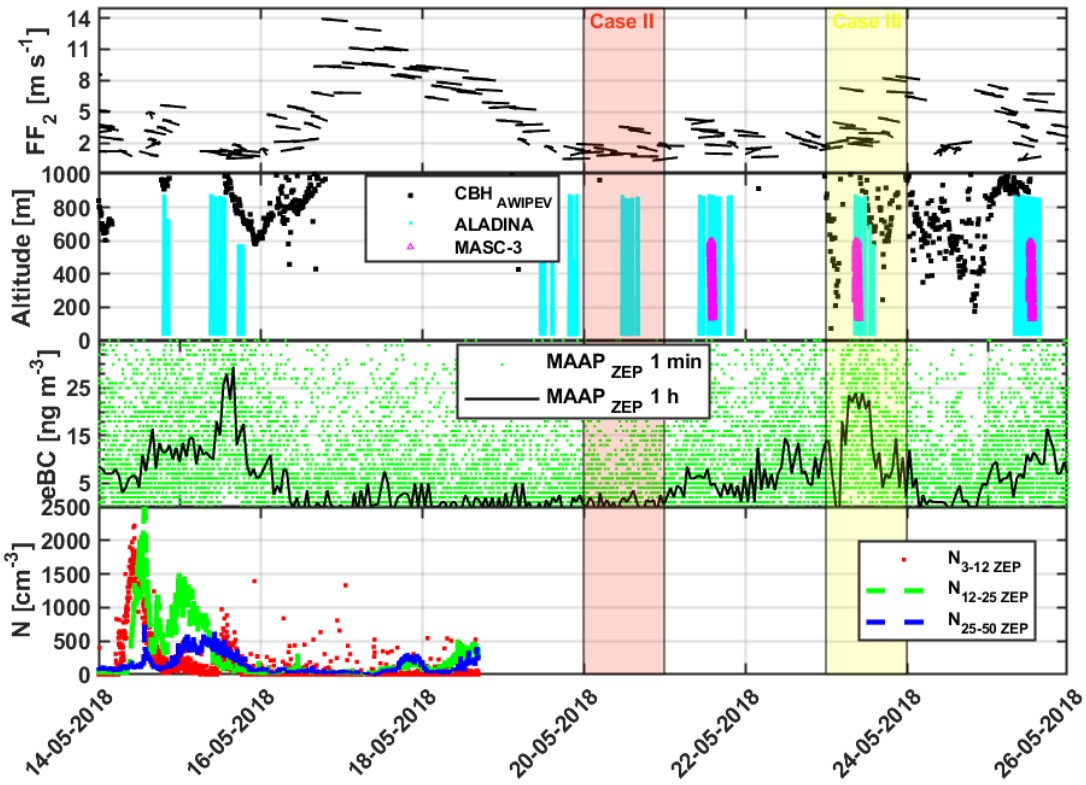

**Figure 4.** The same as Fig. 3 but for the second measurement period of ALADINA between 00:00 UTC on 14 April 2018 and 00:00 UTC on 26 May 2018. The red shading indicates the time series of the second case study analysed here (Case II, Sect. 3.4) that focuses on the horizontal distribution of $N_{3-12}$ and the yellow shading stands for the period of the third case study (Case III, Sect. 3.5) that considers eBC measurements of the AE51 onboard ALADINA calculated from horizontal legs that were performed at similar heights with the Zeppelin Observatory.

Within the flight duration of 1.5 to 2 h, approximately 40–50 legs are sampled that allow to calculate vertical profiles of the mean 3-D wind vector, temperature and humidity.

The first week of the flight campaign was mainly used for unpacking, preparation and test flights of both UAS. In consequence of this, the majority of the research flights was carried out during May 2018 in a transition period between spring and early summer, thus influenced by snow melt, which can be further seen in the reduced snow covered surfaces (Fig. 2). For safety reasons, the field application was limited to operation out of clouds, without precipitation and for wind speed below $15\,\mathrm{m\,s^{-1}}$. Thus, a continuous flight program was not possible during the entire field period, which will be explained in more details in the following.





For additional background information and in order to enable a better orientation of the temporal availability of the data that is used in this study, Fig. 3 and Fig. 4 display specifically chosen measurement parameters derived from different ground based stations (e.g. wind speed, wind direction, cloud base height, eBC, and $N_{3-12}$), separated into two main episodes within the applied flight campaign. More precisely, the first section shows observations between 24 April 2018 and 2 May 2018, and the second part presents data from 14 May 2018 until 26 May 2018. Figure 3a and Fig. 4a show time series of the measured wind speed and wind direction, observed at the AWIPEV research site at the height of 2 m (Maturilli, 2018a, b) in the village of Ny-Ålesund. In addition, time series of ceiling in terms of cloud base height, measured at the AWIPEV station (Maturilli, 2018c, d), are presented in Fig. 3b and Fig. 4b together with the measurement periods of both UAS (see Fig. 3b and Fig. 4b). During the periods from 27–30 April 2018 due to heavy snowfall and from 2 May 2018 to May 13 2018 no measurements were performed due to technical reasons (Fig. 3b). Further, in the presence of low level clouds and high wind speed, no field activity was carried out on 16–18 May 2018 and on 24 May 2018 (Fig. 4b).

In addition, the time series of different aerosol properties are presented in Figs. 3–4 to assure the clarity of the decision for the three selected case studies. Observations of eBC calculated from MAAP at the Zeppelin Observatory are shown for the ALADINA flight period (Fig. 3c and Fig. 4c), as well as nano-SMPS data for three different sizes of $N_{3-12}$, $N_{12-25}$ and $N_{25-50}$ in Fig. 3d and in Fig. 4d. The first case study investigates the period of the end of the Arctic haze between 24 April 2018 and 26 April 2018 (Case I, Sect. 3.3, Fig. 3). The second case study focuses on the horizontal distribution of UFP observed during a day when nucleation occurred at the site on 20 May 2018 (Case II, Sect. 3.4, Fig. 4). In order to discuss the impact of local pollution on the spatial distribution of UFP, a day with a higher degree of local pollution was chosen that can be further seen by the increase of the measured eBC from 0 to a maximum of 24 ng m$^{-3}$ on 23 May 2018 (Case III, Sect. 3.5, Fig. 4).

However, the main focus of the ALADINA investigation is on linking observations of aerosol properties at the different observatories located at different altitudes. Subsequently, the discussions of the results start with an overview of the measured vertical profiles of UFP in the size range of 3 to 12 nm that were performed with ALADINA in connection to the aerosol size distribution measured at the two observations GRU and ZEP.

## 3 Results and discussions

### 3.1 Overview of the vertical variability of aerosol particles during the flight campaign

Figures 5–6 display time series of the total aerosol particle number concentration measured with SMPS at GRU and DMPS at ZEP in comparison with vertical profiles of $N_{3-12}$ based on ALADINA (background), separated into two episodes valid for the first part between 24 April 2018 and 2 May 2018, and for the second part that considers observations from 14 May 2018 until 26 May 2018, thus within the same time slots as presented in Sect. 2.4. At the beginning of the campaign, accumulation mode particles were dominant at both sites with low number concentrations of a few 100 cm$^{-3}$ (Fig. 5), most likely linked to the end of the Arctic haze period. Nucleation mode particles were not present at both sites, but sporadic occurrences of UFP with short term duration and no further growth of the particles can be identified, most apparent at both sites in the evening





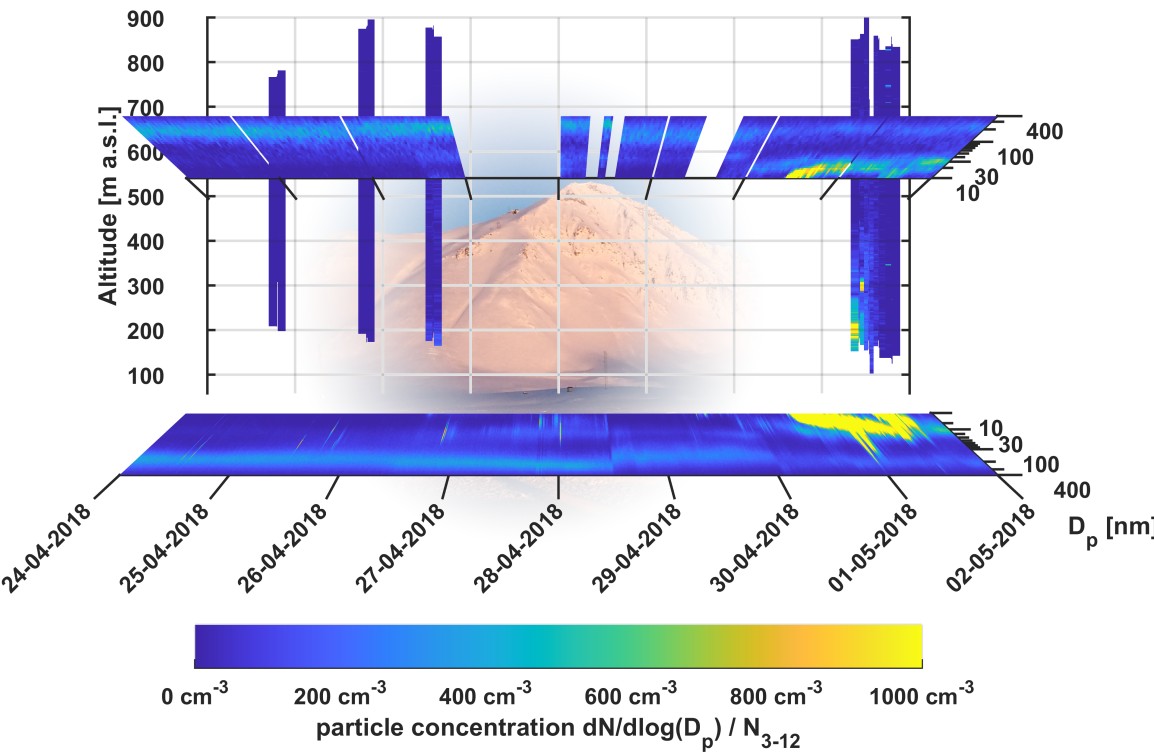

**Figure 5.** Spatial distribution of aerosol particles observed at Ny-Ålesund during the first part of the ALADINA period between 00:00 UTC on 24 April 2018 and 00:00 UTC on 2 May 2018. In-situ size distribution data is presented for a size range between 10 and 400 nm, measured with a DMPS at the Zeppelin Observatory (top) and with a SMPS at Gruvebadet (bottom). ALADINA was operated on four measurement days and performed vertical profiles up to a typical maximum height of 850 m a.s.l. so that profiles of $N_{3-12}$ are shown as a projection between both stations in accordance to the same time series. In order to provide a better comparison, the same colour bar is used for the different observations of the aerosol particle number concentration, ranging from 0 (blue) to 1000 cm $^{-3}$ (yellow). The DMPS data at the Zeppelin Observatory was not available for the whole investigation period, which is characterized by data gaps that occurred from 00:00 UTC on 27 April until 00:00 UTC on 28 April 2018 and temporarily between 28 April and 29 April 2018 which is however out of the ALADINA period. The picture in the background shows the Mount Zeppelin with the Zeppelin Observatory at the height of 472 m a.s.l. and in the low area of the picture is the Climate Change Tower that is deployed at ground level near the Gruvebadet Observatory. The picture was taken at the airport at the beginning of the campaign.

hours on 26 April 2018. The vertical profiles of $N_{3-12}$ show a similar picture by means of no appearance of UFP in the vertical scale, except for a low enhancement of $N_{3-12}$ with around 300 cm$^{-3}$ on 26 April 2018.

After 30 April 2018, accumulation mode particles played a minor role and nucleation appeared at both sites, but with 290  discrepancies in the measured maximum of the number concentrations. As the measured number concentrations are higher at



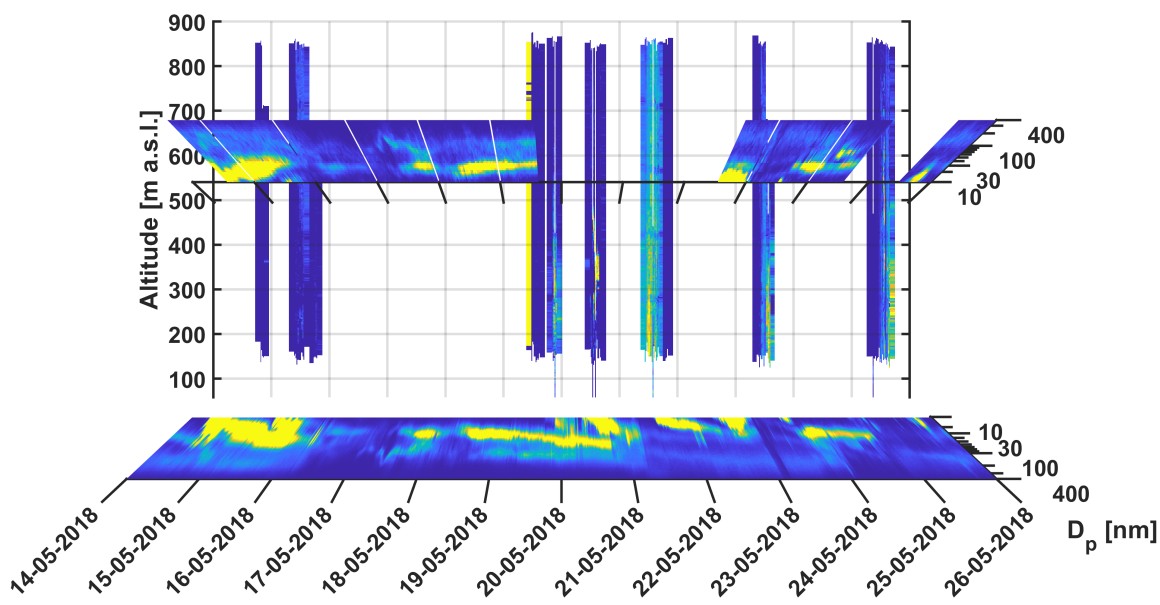

**Figure 6.** The same parameters as shown in Fig. 5 but for the second episode of the ALADINA campaign, which lasted between 00:00 UTC on 14 May 2018 and 00:00 UTC on 26 May 2018. ALADINA was operated on seven days and data gaps of DMPS data at the Zeppelin Observatory were present from 15:00 UTC on 19 May 2018 until 13:00 UTC on 22 May 2018 as well as between 14:00 UTC on 24 May 2018 and 11:00 UTC on 25 May 2018, mainly within the ALADINA period.

GRU, an origin for UFP is possibly connected to a local source near ground. At around midday on 1 May 2018, the subsequent growth of the particles stopped and the vertical profiles of $N_{3-12}$ present a clear domination of UFP close to ground and less particles above the altitude of 280 m a.s.l., thus supporting the idea of a potential hot spot of precursor gases or UFP coming from the surface, that were lifted upwards but prevented from mixing within the whole investigation altitude. During

the second part of the field period (Fig. 6), the nucleation mode was significantly enhanced at both sites. However, only a small degree of new particle formation events with the typical growth of particle size with time, called class I in the classification of Kulmala et al. (2012), could be identified during the period, valid for the ALADINA measurement days on 14 May 2018 and 21 May 2018. One explanation for the high occurrence of inhomogeneous particle growth may be related to rapid changes of air masses that occur frequently at the research area, mainly impacted by the complex terrain. Fast changes of air masses

were dominant in May 2018, most pronounced during the ALADINA observation days on 1 May 2018, 15 May 2018, 19 May 2018 and 25 May 2018, which is evident from rapid shifts of the ground based wind direction and increased wind speed from the AWIPEV station (see Fig. 3a–b and Fig. 4a–b) and by discontinuities in the observed cloud base height. Considering





the vertical profiles of $N_{3-12}$, which are further displayed in Fig. A3, high discrepancies are visible between ground based observations and measurements at the higher altitude range valid for ZEP. This is of particular relevance for the observations

on 19–20 May 2018, a period when nucleation was visible, and for the measurement day on 23 May 2018, where in both cases highest number concentrations of $N_{3-12}$ occurred below the height of ZEP. This demonstrates the pronounced impact of the ABL stability on the vertical mixing of UFP, so that possibly sources of new particle formation from the ground were prevented from mixing within the upper parts of the ABL, thus occurrence of new particle formation cannot be identified by solely taking into account observations at GRU and ZEP. However, both cases strongly differ from the observations on 21 May

2018, where a NPF event of class I was observed at GRU and the vertical profiles of $N_{3-12}$ show an appearance of UFP in the whole studied altitude range up to a maximum altitude of 850 m a.s.l., thus far exceeding the Zeppelin Observatory, so that the nucleation event most likely reached the FT as well, in any case the nucleation event of this class influenced the overall column investigated here.

Summarizing the observations during the presented 22 measurement days in Figs. 5–6, UFP occurred frequently on 55%

of the 12 different measurement days, but the appearances of UFP are mainly linked to non-defined NPF events, thus might not have been assessed after the typical classification for NPF events. Only three NPF events may have been classified as NPF event with subsequent growth rate which further results in a so called "banana-shape" (Heintzenberg et al., 2007). However, for most of the events, the particles' growth was interrupted and lasted until around midday of the following day, for instance during the observations on 30 April-1 May 2018, 14-15 May 2018 as well as on 21–22 May 2018. By considering only well

known classic NPF event days, the frequency of occurrence is significantly reduced to a value of 23%, as the classification is only applicable for five measurement days, which, however, coincides with the study of Lee et al. (2020) who considered a two year data set. Looking at the time series of the vertical profiles (for additional information see Figs. A1–A3), only four out of the total eleven measurement days with ALADINA do not show any occurrence of UFP in the size of $N_{3-12}$, which can be explained by the following: The first three profiles were performed in April, when UFP were not visible at both sites or

solely apparent with a concentrations of a few $100\,\mathrm{cm}^{-3}$ as well as on a short temporal scale which is equally according to the sporadic appearances of $N_{3-12}$ measured with nano-SMPS at ZEP, shown in Fig. 3d. On 14 May 2018, the aerosol particles have most likely reached larger sizes above 12 nm in consequence of a subsequent growth rate of the particles, so that they are out of the size range presented here.

## 3.2   Summary of the vertical distribution of aerosol particles and ABL properties measured with ALADINA

Figure 7 presents a statistical analysis (median, 25% and 75%, maximum) based on histograms which comprise all 230 vertical profiles that were performed with ALADINA during the period. More precisely, the histograms are based on vertical profiles of aerosol particle number concentration in different sizes $N_{3-12}$, $N_{>12}$, $N_{300-500}$, potential temperature $\theta$ and water vapour mixing ratio $r$ between a typical height of 150–850 m a.s.l.. Thus, this altitude area excludes surface measurements with AL-ADINA and due to safety reasons, the majority of the profiles started at an altitude of 100 m above ground level (a.g.l.) and as

the airport is located at a level of around 40 m a.s.l., all profiles are bordered in the specific altitude above 150 m a.s.l. in order to provide the highest statistical relevance. The vertical distribution of $N_{3-12}$ shows a higher concentration close to ground



with decreasing number concentrations with increasing altitude. The median of $N_{3-12}$ is low between 90 and 270 cm$^{-3}$ with an overall minimum at the height of 550 m a.s.l., suggesting a generally low frequency of UFP above ZEP. However, the total maximum of $N_{3-12}$ exceeds 6,200 cm$^{-3}$ at the height of 640 m a.s.l., thus the highest number concentrations were found even

above the height of the Zeppelin Observatory. Here it is important to note that the maximum is not shown in the graph in order to fulfil the readability of the vertical distribution of $N_{3-12}$, as the maximum was far out of the 75% range as well. The vertical profile of particles with a particle size larger than 12 nm ($N_{>12}$) displays equally higher number concentrations at ground and decreasing values with growing altitude. The median of $N_{>12}$ varies between 420 and 950 cm$^{-3}$ for the entire altitude range. In addition, the total maximum of 14,500 cm$^{-3}$ was measured at the height of 800 m a.s.l. but to a major part the highest number

concentrations appear below 330 m a.s.l. and are associated with strongly variable number concentrations ranging from 1,320 to 13,000 cm$^{-3}$.

Considering the vertical distribution of particles larger than 300 nm ($N_{300-500}$), only several particles cm$^{-3}$ were detected during the period, meaning less than 7 cm$^{-3}$ for the interquartile of 75%. Again, the maximum is not included in the graph in consequence of the same reason as explained for the vertical profiles of $N_{3-12}$, as it is far out of the measurement area represented by the interquartile of 75%. After subsequent nucleation, valid for the measurement days on 14 May 2018, 15

May 2018, 19 May 2018, 23 May 2018 and 25 May 2018, UFP grew to larger sizes and were recorded by the OPC. For instance, a total maximum of 120 cm$^{-3}$ occurs in the whole altitude after the NPF event on 14–15 May 2018. In general, the highest number concentrations were measured during April 2018 due to the main presence of accumulation mode particles. The vertical distribution of the water vapour mixing ratio $r$ indicates an influence of maritime air masses with enhanced moisture close to ground and dryer air lifted above. The median of $r$ decreases from 2.6 g kg$^{-1}$ at 150 m a.s.l. to 2.2 g kg$^{-1}$ at the height

of 850 m a.s.l., and the total maximum of 3.7 g kg$^{-1}$ was measured on 15 May 2018, when the cloud base height reached low altitudes of 600 m a.s.l. (see Fig. 4) so that the UAS was not operated as high as usual in order to assure a safe mission. The vertical profiles of $\theta$ show a higher variability in the vertical scale, ranging from stable conditions in the 75% line below the height of 400 m a.s.l. and a generally well mixed stratification in respect of the median of $\theta$ that represents a marginal deviation

of 0.33 K in the whole altitude range between 150 and 850 m a.s.l., thus leading to the assumption of a high potential of mixing of UFP within the ABL.

In contrast to the summary that takes into account all vertical profiles (Fig. 7), Fig. 8 depicts the same selected parameters, but under the requirements that solely vertical profiles are considered as histograms when UFP are detectable at both research sites and the difference of both CPCs onboard ALADINA passes the total concentration of 500 cm$^{-3}$. These criteria were

chosen in order to avoid any likely impact of artefacts on the appearance of UFP. The vertical profiles of $N_{3-12}$ show a similar distribution, by means of a general decline of number concentration with growing altitude. However, a higher variability is visible in the vertical between the heights of 150 and 550 m a.s.l., in contrast to the summary when all vertical profiles are considered for the analysis. The same effect is obvious in the vertical profiles of particles with a size larger than 12 nm, indicating a general decrease of the number concentrations with altitude, but with higher gradients below the height of 550 m a.s.l., thus

mostly in agreement with the vertical pattern of $N_{3-12}$. The vertical distribution of $N_{300-500}$ shows minimal number concentrations below 10 cm$^{-3}$ in the whole altitude range. In comparison with all profiles, the maximum of $N_{300-500}$ is visible in

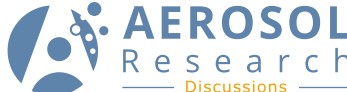

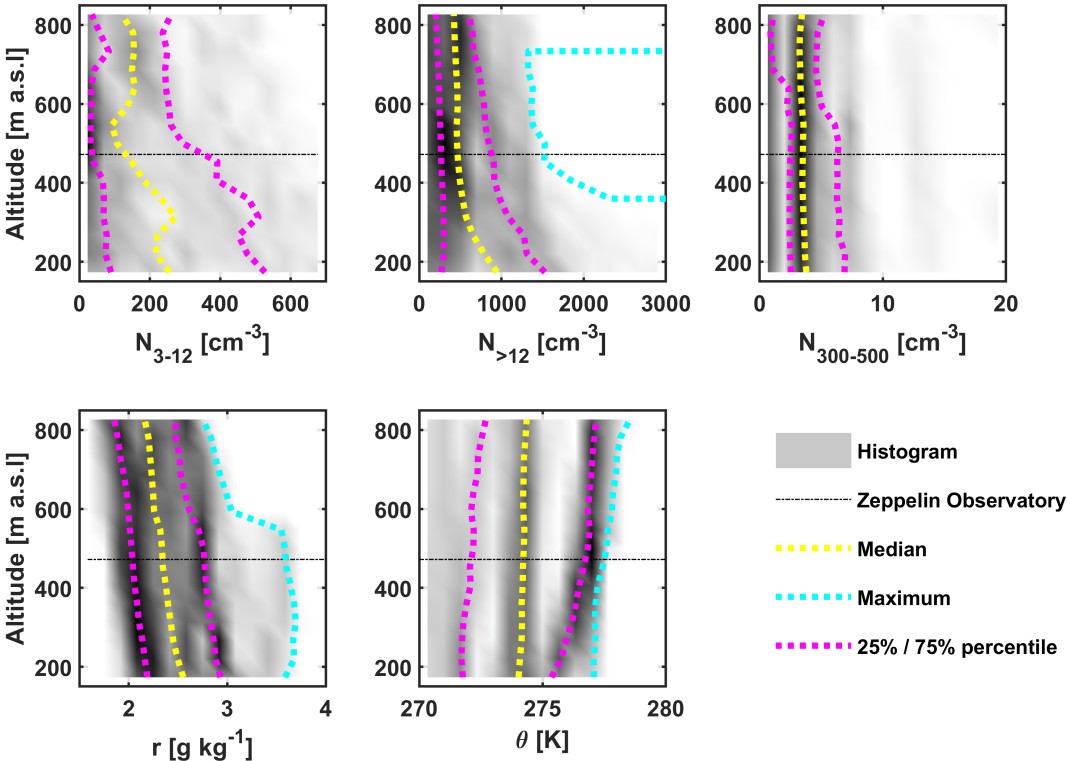

**Figure 7.** This figure shows a summary of all vertical profiles of selected parameters that were measured with ALADINA between the height of 150 and 850 m a.s.l. in Ny-Ålesund from 24 April until 25 May 2018. The presented histograms are colour coded in grey and based on 230 vertical profiles that are individually presented in a temporal sequence in Figs. A1–A6. From left to right: aerosol particle number concentration between the size of 3 and 12 nm $N_{3-12}$ in cm$^{-3}$, aerosol particle number concentration with a size of larger than 12 nm $N_{>12}$ in cm$^{-3}$, and between 300 and 500 nm $N_{300-500}$ in cm$^{-3}$, water vapour mixing ratio $r$ in g kg$^{-1}$ and potential temperature $\theta$ in K. The yellow line represents the calculated median of all profiles, the cyan line stands for the specific maximum and the bright magenta lines mark the measurement range between 25% and 75%, respectively. The maxima of $N_{3-12}$ and $N_{300-500}$ are not provided in the graph in order to provide a better readability of the analysis, as they are far outside of the measurement range. The black dashed line indicates the height of the Zeppelin Observatory.

this graph, as it is significantly reduced to a minimum of 9 cm$^{-3}$ at the height of 360 m a.s.l., and a maximum of 27 cm$^{-3}$ at the lowest calculated height of 150 m a.s.l.

The vertical distribution of the water vapour mixing ratio $r$ indicates an impact of dryer air masses on the enhanced appearance of UFP, as the median is reduced in comparison with all profiles, by ranging between 2.1 and 2.4 g kg$^{-1}$. In addition, the maximum of $r$ decreases as well to 2.6–3.2 g kg$^{-1}$, which is in agreement with UFP that occur more frequently during



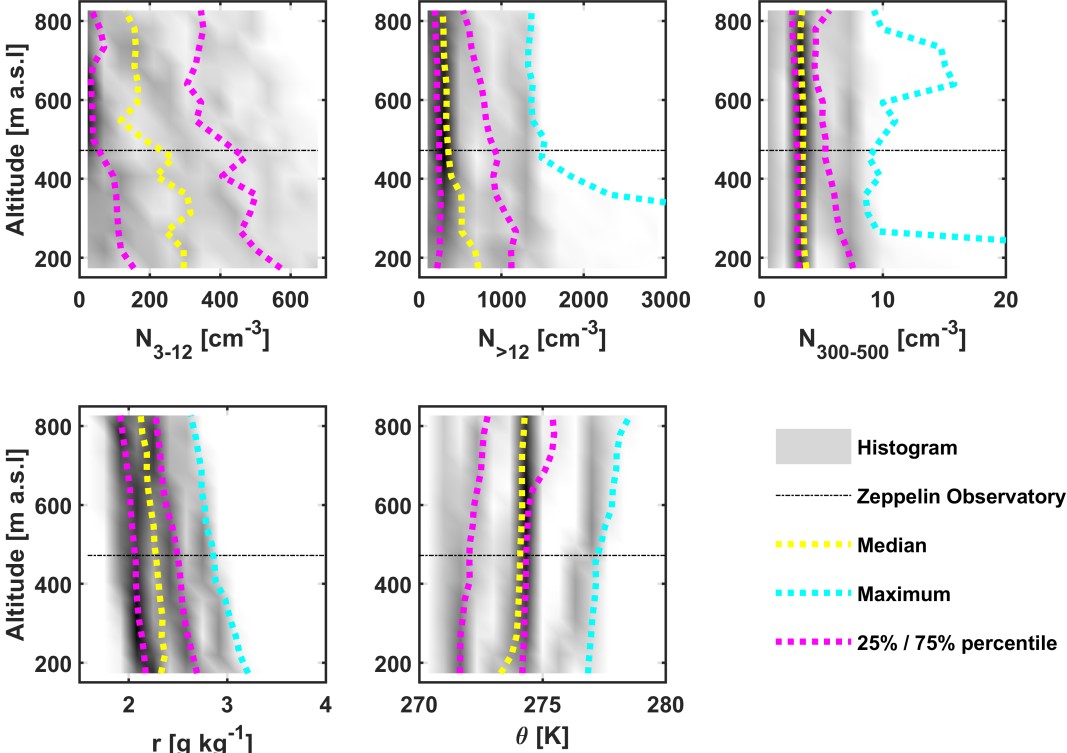

**Figure 8.** The same as discussed in Fig. 7, but with a different chosen criterion for the statistical analysis. Vertical profiles of the parameters are solely considered when they are subject to the condition that UFP were apparent at both ground based stations and $N_{3-12}$ exceeds values of $500\,\mathrm{cm^{-3}}$. The choice was made in order to exclude observations that may correlate to artefacts of the CPCs measurement range.

cloud free phases. The vertical profiles of the potential temperature $\theta$ demonstrate stronger gradients of the ABL for the chosen criteria. The median vertical profile of $\theta$ displays a generally well mixed layer except for a pronounced inversion layer in the lowermost $300\,\mathrm{m\,a.s.l.}$ that further coincides with highest measured UFP concentration and a general marginal accumulation
mode at the same altitude.

In principal, and valid for both situations, the vertical distribution of UFP shows higher number concentrations close to ground. This corresponds with enhanced moisture near the surface, which can be explained by the fact that the site is directly situated at the coast, which may imply a high potential for local water vapour and other precursor sources originating from the sea. One of the major dominant sources for the measured UFP may be linked to MSA as precursor, as recently shown in Beck
et al. (2021). In addition, according to the vertical profiles, UFP occur at the Zeppelin Observatory and even above, meaning that NPF may achieve larger spatial scales and may even exist within the FT. The summary shows a clear impact of the ABL stability on the vertical distribution of UFP, as the vertical profiles of $\theta$ significantly differ by means of a well mixed ABL



(median) for all selected vertical profiles, and more inversion layers are present for taking into account only the NPF days with the chosen NPF criterion. Interestingly, large gradients of UFP occur in the vertical distribution in both cases in the lowermost

550 m a.s.l, even when a generally well mixed ABL is apparent. This implies that additional effects, most likely linked to wind shear due to the complex topography contribute to the high variability of UFP in the vertical scale. All vertical profiles in Figs. 7–8 indicate discrepancies of the measured parameters between the two ground based stations GRU and ZEP, so that the question arises which of the research sites might be the most representative one for aerosol long term monitoring at an Arctic coastal site.

Here it is important to note that these two graphs were chosen for a general overview and the results are solely based on observations during the eleven measurement days with ALADINA, when the weather conditions allowed a safe field operation with the UAS. Thus, and based on the fact that a high difference exists of observations between the two ground based stations, further case studies are discussed in the following sections in more detail in order to (1) better understand the impact of the ABL stability on the aerosol distribution in the vertical scale, (2) to access a likely influence of horizontal effects like wind shear

and local sources that may better explain discrepancies at the two different ground based observations, and (3) to demonstrate the capabilities of ALADINA, that enables additional investigations like studying the amount of accumulation mode particles with the integrated OPC. Fur this purpose, specific selected case studies are shown. Case I considers selected vertical profiles of aerosol particles during the end of the Arctic haze period on 24–26 April 2018 (see Sect. 3.3). Case II takes into account horizontal observations of $N_{3-12}$ on 20 May 2018, where a persistent NPF event was measured at ground (see Sect. 3.4) and

Case III on 23 May 2018 represents a day with ship activity at the port and enhanced local traffic (see Sect. 3.5).

### 3.3    Case I: Sporadic appearance of UFP during the end of Arctic haze influenced by onshore wind on 26 April 2018

During the first part of the field experiment in April 2018, the aerosol composition was affected by Arctic haze in Ny-Ålesund, thus influenced by phenomena on regional scales. This can be supported by a clearly enhanced accumulation mode that was apparent at both research sites, as presented in Fig. 5, and given by the fact of similar measured number concentrations presented

for other Arctic research sites like Alert station in Canada (Abbatt et al., 2019) or at Villum Research Station in Greenland, as shown in Nguyen et al. (2016) and Dall'Osto et al. (2019) during measurements in April of different years. In addition, the 1 h averaged eBC calculated from MAAP as an indicator of air pollution (Fig. 3c) reached highest values up to 60 ng m$^{-3}$ during the period of 2–30 April 2018, and then decreased to 5–22 ng m$^{-3}$, thus Arctic haze was no longer apparent at the site.

Figure 9 displays exemplarily four selected vertical profiles of $\theta$, $r$, $N_{>3}$ and $N_{300-500}$ measured with ALADINA between

the height of 0–900 m a.s.l. at 18:45 UTC on 24 April 2018, at 19:20 UTC on 25 April 2018, at 13:30 UTC and at 13:58 UTC on 26 April 2018 2018. In addition, the calculated horizontal wind direction $dd$ is presented based on two research flights that were performed with MASC-3 which started at 20:00 UTC on 24 April 2018 and at 12:52 UTC on 25 April 2018. For a better orientation, the heights of GRU and ZEP are indicated in the figure as well.

The ABL shows multilayer structures, which are visible in all four different vertical profiles of $\theta$, but most pronounced during

the second profile on 25 April 2018, where two distinguished inversion layers are present at the two height levels of 300 and 600 m a.s.l., respectively. In general, the water vapour mixing ratio $r$ reached marginal values, but the effect of local maritime





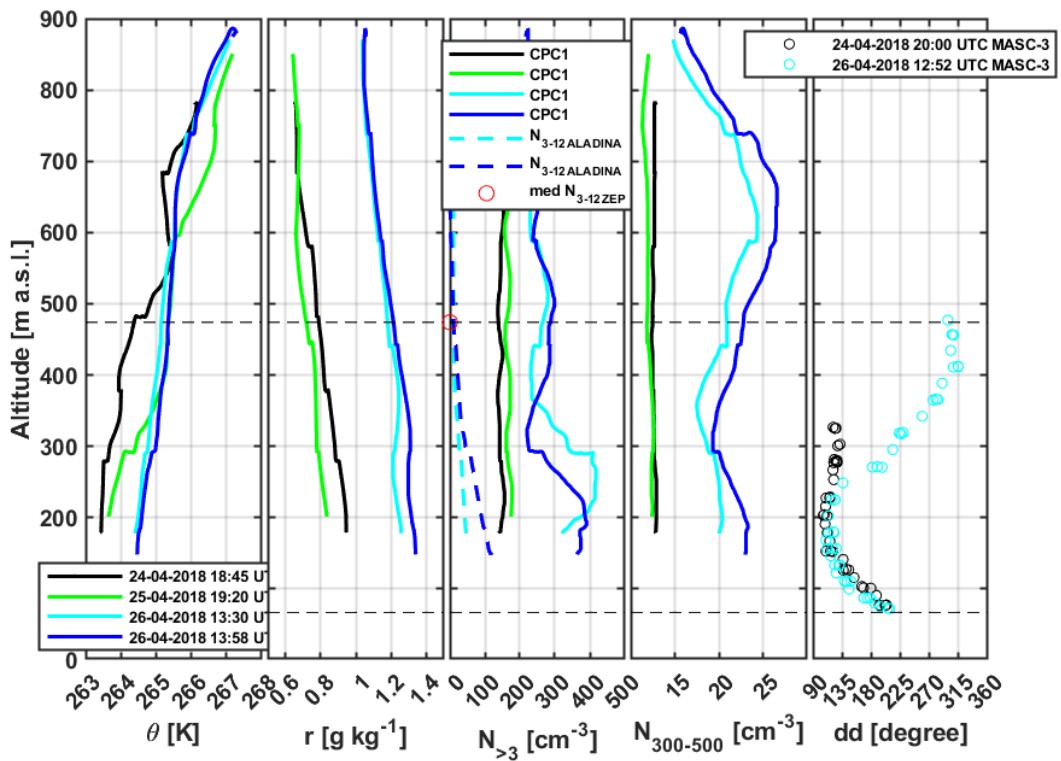

**Figure 9.** Case I: Selected vertical profiles measured with the two UAS ALADINA and MASC-3 during between 24 April and 26 April 2018. From left to right: potential temperature $\theta$ in K, water vapour mixing ratio $r$ in $\mathrm{g\,kg^{-1}}$, aerosol particle number concentration larger than 3 nm $N_{>3}$ in $\mathrm{cm^{-3}}$, aerosol particle number concentration larger than 300 nm $N_{300-500}$ in $\mathrm{cm^{-3}}$ measured with ALADINA at 18:45 UTC on 24 April 2018 (black line), at 19:20 UTC on 25 April 2018 (green line), at 13:30 UTC (cyan line) as well as at 13:58 UTC on 26 April 2018. The horizontal wind direction $dd$ in degree was estimated along with measurement flights of MASC-3, after take-off at 20:00 UTC on 24 April 2018 (black circle) and at 12:52 UTC on 26 April 2018. The two black dashed lines stand for the height of the Zeppelin Observatory (top) and Gruvebadet (bottom). For a comparison, vertical profiles of $N_{3-12}$ in $\mathrm{cm^{-3}}$ are added to the third graph, based on measurements with ALADINA at 13:30 UTC (cyan dashed line) and at 13:58 UTC on 26 April 2018 (blue dashed line) as well as the median concentration that was calculated from nano-SMPS data at ZEP (red circle), averaged within the corresponding ALADINA measurement profiles.

air, which was advected from the coast, is visible to a small degree in the vertical distribution in terms of enhanced values of $r$ close to ground, which decreases with altitude. The aerosol particle number concentration measured with the CPC1 in the size between 3 and 1 $\mu$m ($N_{>3}$) shows two main layers in the vertical scale. Low number concentrations of around $150\,\mathrm{cm^{-3}}$ were observed on 24 April 2018 as well as on 25 April 2018. In contrast to those observations, enhanced number concentrations were visible in the lowermost 360 m a.s.l. with a maximum of $400\,\mathrm{cm^{-3}}$ at 13:30 UTC on 26 April 2018. $N_{3-12}$ were not detectable






with ALADINA between 24 April 2018 and 25 April 2018, but occurred to a small degree below the specified inversion layer at around 300 m a.s.l. and were enhanced in lower altitudes reaching maxima of 50 and 120 cm$^{-3}$ at the height of 150 m a.s.l. on 26 April 2018, assuming a weak local source for UFP that originated from the surface. However, in consequence of the
existence of the inversion layer, mixing was suppressed. The general low appearance of $N_{3-12}$ coincides with the observations of the nano-SMPS (Fig. 3d) in terms of the same particle size and for the measurement period. However, the situation changed during the day, when $N_{3-12}$ occurred more frequently, but still on a sporadic pace along with low level clouds and wind direction from SW at ground, but disappeared completely during midnight. According to the vertical patterns of wind direction $dd$, wind shear is visible on 24 April 2018 and 25 April 2018, changing from SE to E between the height of GRU and the
height of 200 m a.s.l., and four wind regimes existed on 26 April 2018, further influenced by a shift from E to NNW within the altitude range between GRU and ZEP. Projecting the current wind direction to the topography, the calculated horizontal wind indicates in the vertical scale an origin from the Zeppelin Mountain between the surface and up to the height of around 150–200 m a.s.l., where the wind direction merged to onshore wind with a wind direction of SE and this wind regime coincided with higher number concentrations of $N_{3-12}$. Between the height of 250 and 400 m a.s.l., the wind turned to offshore wind, in
accordance with a decrease of $N_{3-12}$ in the vertical pattern. Above 400 m a.s.l., the fourth wind regime was identified which originated from the water, but upwards from the fjord in NW along with an enhancement of $N_{300-500}$, which is, however, only lifted upwards from the inversion layer in the higher altitude region, leading to the assumption of a high degree of sea salt aerosol that was measured within the particle size of 300 to 500 nm.

To sum up the findings based on the vertical profiles shown here, the vertical distribution of aerosol particles was strongly
connected to ABL properties. In particular, gradients with enhanced and locally confined concentrations were linked to the ABL stability and significantly affected by the current wind field. In addition, UFP tended to occur during the end phase of Arctic haze with only low concentrations, and solely sporadically on short temporal (see Fig. 3 and Fig. 4) and without any subsequent growth of the particles. This could be related to the existence of the high pre-population of larger particles that suppressed NPF most likely due to altered polluted emissions that were transported to the site. However, this case study considers observations
of number concentrations with a few 100 cm$^{-3}$ and lower, thus the aerosol sensors of ALADINA worked on their detection limits. The low UFP concentrations are confirmed by the UFP measured sporadically at ZEP (see Fig. 3d) as well.

### 3.4 Case II: High variability of the horizontal distribution of UFP observed during nucleation on 20 May 2018

Figure 10 shows the horizontal distribution of $N_{3-12}$ along legs that were performed at three different constant altitudes (from left to right) during four measurement flights (from top to bottom) with ALADINA between a period from 11:44 to 14:34 UTC
on 20 May 2018. Each flight pattern consists of legs that cross the coast in direction from the airport to the sea, a full operation above open water by heading to SE with a distance of around 2 km. The turnaround from the sea back to the airfield is used for achieving the next altitude level, thus this part is not considered for the study as the variability may be attributed to changes in the altitude.

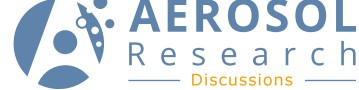



**Figure 10.** Case II: Horizontal distribution of $N_{3-12}$ observed during four (out of five) research flights that were carried out with ALADINA on 20 May 2018. The horizontal legs are operated in three different altitude levels, ranging from 173–192 m a.s.l. (left), 314–334 m a.s.l. (middle) as well as 458–478 m a.s.l. (right) and the flights are directed from the airport to the sea (SE to NW), above sea heading into E and coming back to the airfield. The last part of the horizontal leg is used for achieving the next defined altitude level, thus is excluded from the analysis. Each studied horizontal leg is indicated in white and the full flight pattern is shown in the background (black dashed line). From top to bottom: $N_{3-12}$ during the first flight between 11:42 and 11:50 UTC, measured during the second flight from 12:35 until 12:43 UTC, during the third flight in the period of 13:36 and 13:44 UTC as well as during the fourth flight that lasted between 14:26 and 14:34 UTC. The colour bar ranges from 0 (blue) to $200\,\mathrm{cm^{-3}}$ (red) for the first and second flight and is enhanced to a maximum of $800\,\mathrm{cm^{-3}}$ for the third and fourth research flight.





This day was chosen for analyses as the research flights were performed continuously when nucleation mode appeared at
GRU with ongoing subsequent growth rate which started at around noon at 12:00 UTC (Fig. 6). At 11:43 UTC, an enhanced
aerosol particle number concentration of $N_{3-12}$ occurred near the coast and above sea at the mean altitude of 192 m a.s.l.
with minimal concentrations of 50–120 particles cm$^{-3}$. Several minutes later, on the same horizontal scale, but at a mean
height of 334 m a.s.l., $N_{3-12}$ reached marginal number concentrations, and no UFP were visible at the higher altitude range
of 478 m a.s.l, which is almost the same height level as Zeppelin. According to the vertical profiles of $\theta$ (Fig. A5), the ABL
was stably stratified below the Zeppelin Mountain, so that a mixing of particles up to the FT was not possible. This implies
that the occurrence of $N_{3-12}$ most likely originates from close to ground with the main source coming from the sea, and a
further mixing is prevented in upper parts of the ABL due to stable conditions. The situation changes during midday, when
$N_{3-12}$ is apparent at lowest altitude of 179 m a.s.l. with highest number concentrations of more than 200 cm$^{-3}$ above the sea.
At 12:40 UTC, $N_{3-12}$ disappears at the height of around 320 m a.s.l. and arises with high variability of the measured number
concentrations at the height of 464 m a.s.l. with pronounced concentrations above the open water. In the afternoon at 13:36
UTC, only a few particles were detected at the height level of 173 m a.s.l. but the number concentration increased significantly
to more than 800 cm$^{-3}$ at the upper height of 314 m a.s.l. at 13:40 UTC, but UFP disappeared at the height of 458 m a.s.l.
Interestingly, the spatial distribution of UFP is similar almost 1 h later, but the total number concentration shows a higher
variability in the horizontal scale at the height of 321 m a.s.l., indicating either a transport of UFP coming from the coast in
direction to the village of Ny-Ålesund or a second local hot spot that initiated the sporadic occurrence of UFP.

In general, the horizontal investigation of $N_{3-12}$ indicates a high variability in the selected altitude regions that could be not
identified by solely taken into account ground based observations. A more frequent appearance of UFP is visible above sea in
comparison with a generally lower measured number concentration above land and close to the airport. However, this was the
opposite during the last research flight on this day, when $N_{3-12}$ showed highest concentration near the village. In addition, it
was verified that $N_{3-12}$ is strongly related to the ABL stability, so that different layers of UFP may have coexisted at specific
altitude levels in consequence of prohibited vertical mixing within the ABL. Rapid changes, like wind shear on small spatial
scale, may indicate a high impact of the topography, so that UFP have been transported to the site, but most likely originated
from outside and existed for longer periods within locally confined vertical altitude ranges.

### 3.5 Case III: Polluted local emissions as a source for UFP on 23 May 2018

This case study considers observations during a day with enhanced local pollution that was emitted at the port by ship and car
traffic in consequence of enhanced logistical activity in comparison with other days when no supply was delivered to the port.
The hypothesis of potential anthropogenic emissions can be verified by the increase of eBC based on the MAAP observations
in the morning hours, shown in Fig. 4. A maximum of 24 ng eBC was measured at around noon and then eBC decreased on a
rapid temporal scale to 10 ng eBC in the afternoon. Interestingly, the enhanced eBC coincided with a sporadic occurrence of
UFP that was measured at both research sites at the same time (see Fig. 6). However, the observed UFP did not grow to larger
particle sizes, instead they disappeared at around afternoon when snow fall was apparent at the measurement site which can be
further seen by the significant decrease of the cloud top base (see Fig. 4). Figure 11 shows the horizontal distribution of $N_{3-12}$

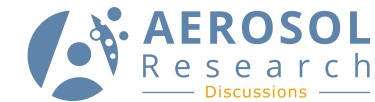

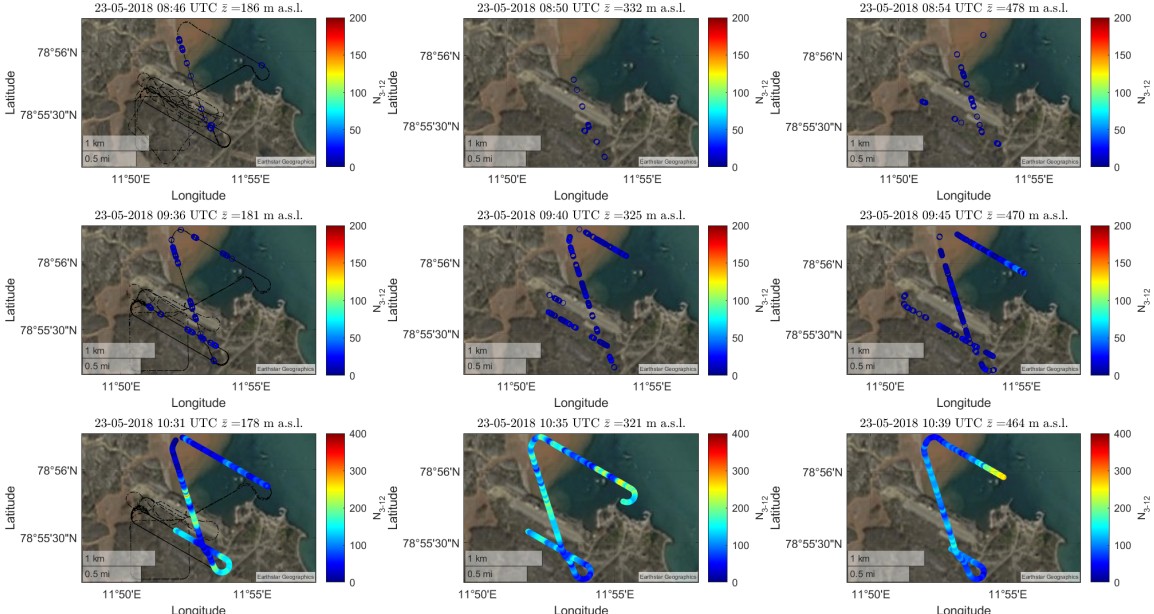

**Figure 11.** Case III: Horizontal distribution of $N_{3-12}$ measured during three (out of six) research flights at three different altitudes between 08:46 and 10:39 UTC on 23 May 2018. The color bar ranges from 0 (blue) to $200\,\mathrm{cm}^{-3}$ as well as up to $400\,\mathrm{cm}^{-3}$ (red) during the last flight presented here.

during the morning hours when the supply was delivered to the site. During the first and second flight, low concentrations of $N_{3-12}$ were measured in all three different altitude levels, but the concentrations increase when the UAS was heading to the village and port. Higher concentrations of $N_{3-12}$ were observed during the third flight, here shown along patterns in three different altitudes between 10:31 and 10:39 UTC. The values exceed $400\,\mathrm{cm}^{-3}$ with the main origin from the village. Thus, UFP may be released from polluted emissions via ship and car traffic at this site.

## 4  Concluding remarks and outlook

The two UAS ALADINA and MASC-3 were applied for studying atmospheric properties and aerosol particle spatial distributions at the research area Ny-Ålesund during melting season between 24 April 2018 and 25 May 2018. In total, 49 research flights were carried out on 11 measurement days with ALADINA for investigating the horizontal and vertical distribution of aerosol particles between ground and up to a maximum height of 850 m a.g.l., which led to 230 vertical profiles during the flight period. MASC-3 was used to analyze the wind field and was operated in parallel during six common measurement days. This article provides an overview of the campaign and the ensemble of flights. The results presented here focus on the vertical distribution of the measured atmospheric parameters of potential temperature, water vapour mixing ratio and aerosol





particles, ranging from nucleation mode of UFP with a size between 3 and 12 nm to accumulation mode with particles larger than 300 nm. The vertical profiles were linked to continuously measured time series of aerosol size distribution derived from the two research sites which are deployed for long term measurements at different altitudes in order to provide a 4-D picture of aerosol properties. In general, high discrepancies of the UFP concentration were observed between the two research sites,
assuming a large impact of the ABL dynamics on the occurrence by means of transport of UFP.

On 26 April 2018 and during the Arctic haze period, the vertical distribution of aerosol particles was significantly affected by wind shear, which mainly results from the complex terrain of the investigation area. With MASC-3, horizontal flight legs were performed near Ny-Ålesund above land and above open water areas from the Kongsfjord in order to link between transport of UFP. Here it is obvious that UFP existed on short period of time and were connected to onshore wind, thus assuming
biological activity from the open water as a main contributor for the origin of UFP. On 19–21 May 2018, the highest number concentrations of $N_{3-12}$ were observed in relation with a persistent inversion layer that existed within the altitude area. Further, the appearance of UFP was a wide spreading event by reaching the whole investigation altitude. However, a clear source cannot be identified, as the formation process has already started during the airborne experiment. In addition, on 23 May 2018, UFP were solely observed below the altitude of the Zeppelin Observatory during a day affected by local traffic, which coincided with
an increase of eBC since morning hours. For validation, the airborne eBC data was compared with ZEP which was, however, not in a good agreement, reaching an overestimation of up to 8 times in comparison with fixed point data at the similar altitude. This in turn is not an artefact, the only reasonable explanation for this is linked to low background aerosol concentration, thus the AE51 was working within the detection limit and is not a feasible tool for operations in a generally clean environment.

To conclude, this study may help to address fundamental open questions based on the feature of the shown spatial distribution
of aerosol particles and the correlation with ABL properties. For instance, at which altitude does NPF take place? However, this question can not be directly answered, as according to the vertical profiles of the measured UFP, a clear typical height could not be identified, as UFP were observed at ground but to a high fraction as well within all studied altitudes. During some event on 1 May 2018, UFP occurred at ZEP before ground, thus a mixture of transport and entrainment might play a dominant role for the appearance of UFP at the measurement site as well. Nevertheless, a trend can be derived that UFP are more enhanced close to
ground, thus leading to the assumption of a high potential of local sources, most likely linked to the open sea, but it cannot be ruled out that sea ice melt was another trigger for NPF as well. Considering the shown horizontal variability of UFP, it seems that UFP are restricted to at least some hot spot but can coexist in different altitude levels as well. Thus, ABL properties play a significant impact on the vertical and horizontal distribution but it can not be ruled out that other sources were simultaneously available during the period, this can not be investigated in detail in case of a restricted investigation area of a few km$^2$. Lee et al.
(2020) studied UFP properties at the Zeppelin Observatory based on a two year data set from October 2016 to December 2018, thus comprising the time period of the UAS application presented here. The UFP occurrence frequency was estimated to 23 %, which is similar to studies at continental sites and was matching the frequency of occurrence during the ALADINA period in case of considering only NPF events of class I. Further, the calculated mean growth rate (GR) for the particle size from 3 to 25 nm was low with 2.66 nm h$^{-1}$, and the mean nucleation rate for the same size range was 0.12 cm$^{-3}$ s$^{-1}$, thus significantly
lower in comparison with other sites in the world (e.g., Nieminen et al., 2018). Interestingly, the study specified high variances





of GR for the size of 3–25 nm with 0.48 to 6.54 nm h$^{-1}$, thus UFP may grow on a rapid pace during some occasions, which is generally not assumed for polar studies. The measured highest values of the GR are similar to rural observations and those high GR were temporarily measured during the ALADINA period as well but a clear connection to ABL processes which may have accelerate the growth of the particles could not be identified.

Altogether, the use of unmanned aerial systems leads to a new opportunities to investigate small scale variability, relate aerosol distributions to local atmospheric dynamics and connect observation sites. Besides process understanding, the data sets are urgently needed for validating high resolution simulations for complex terrain, in order to transfer results to different sites and derive larger scale impact.

**Appendix A:  Time series of vertical profiles of selected measurement parameters based on ALADINA during the**
**investigation period in Ny-Ålesund**

Figures A1–A6 display the time series of the measured vertical profiles at the altitude range of 150 to 850 m a.s.l. of selected parameters during the whole investigation period. The colour bar is indicated in the individual graphs respectively. The authors intended to provide those figures in order to allow a better reproducibility of the outcome of the analyses represented by the normalized histograms, shown in Fig. 7 and Fig. 8.



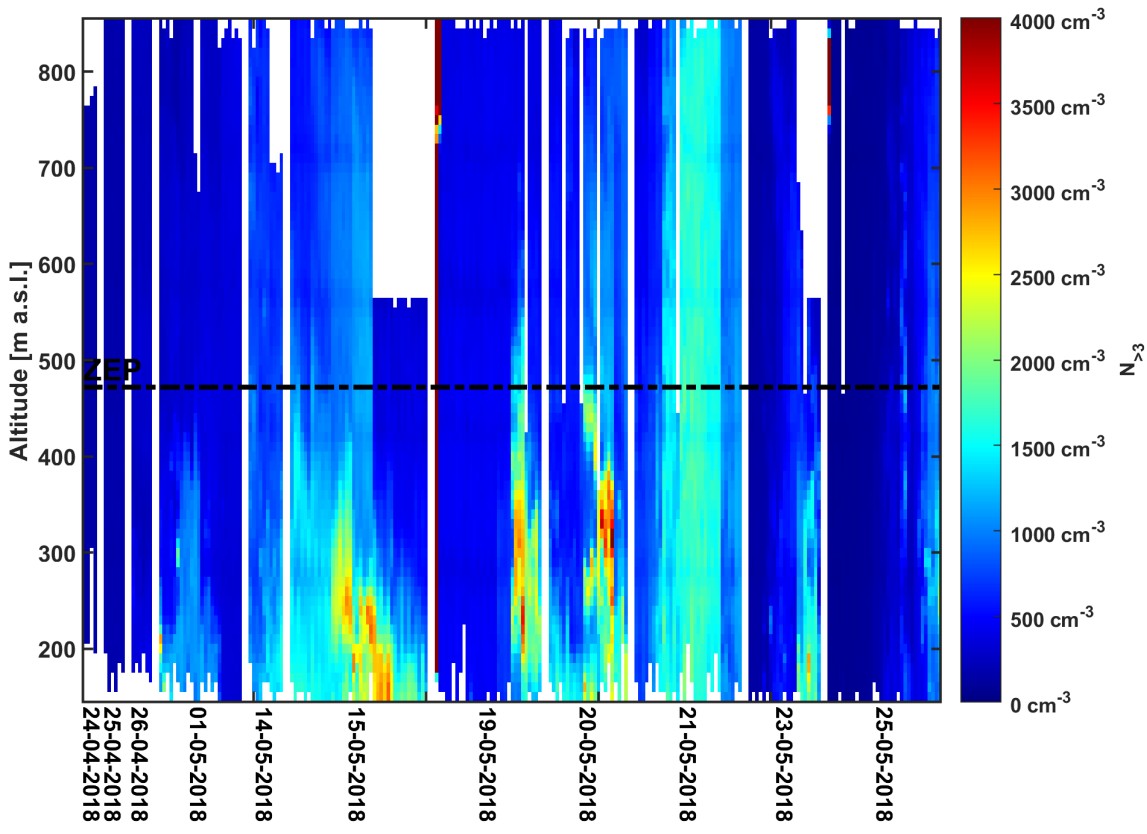

**Figure A1.** Time series of 230 vertical profiles of aerosol particle number concentration measured with CPC1 in $cm^{-3}$ for the size between 3 nm and 1 $\mu$m ($N_{>3}$) on ALADINA in Ny-Ålesund between 24 April and 25 May 2018. The colour bar ranges from 0 $cm^{-3}$ (blue) to 4000 $cm^{-3}$ (red). The dashed black line represents the height of the Zeppelin Observatory (ZEP). Additional information: The analyses presented in Figs. 7–8 are subject to the profiles shown here in terms of normalized histograms.





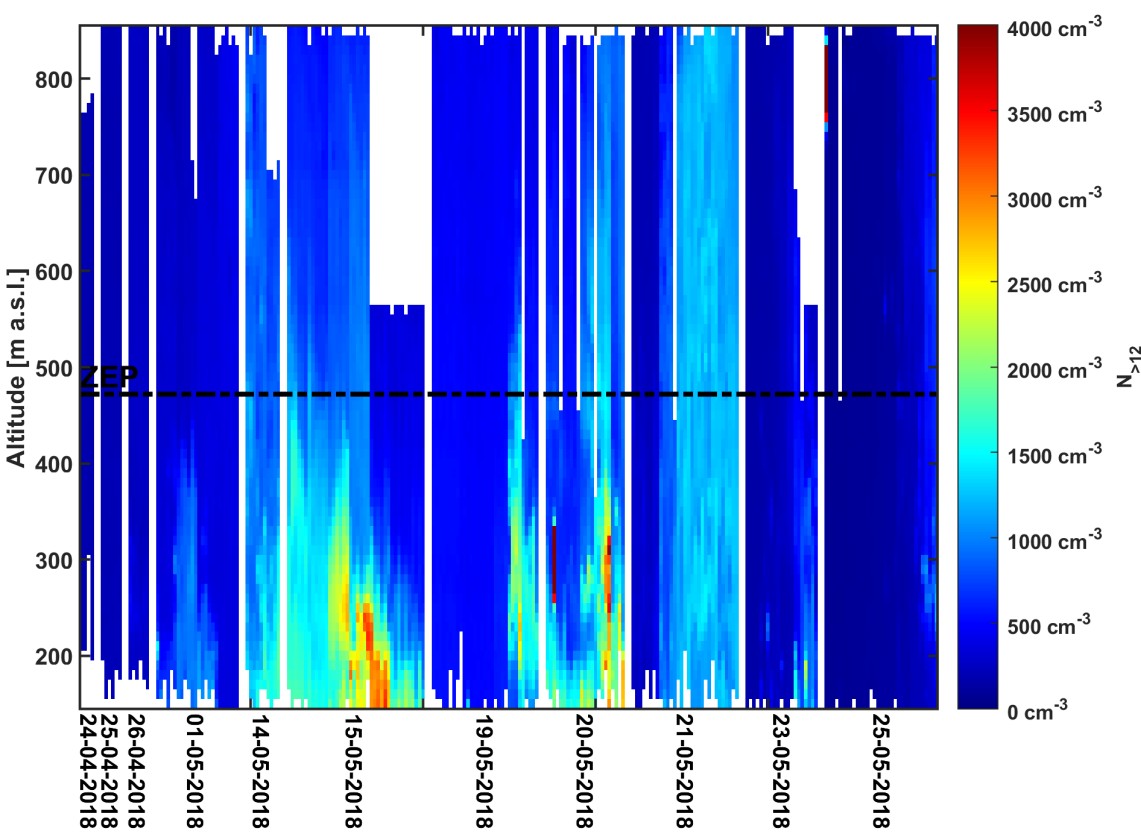

**Figure A2.** The same as Fig. A1, but time series of the vertical profiles of aerosol particle number concentration measured with CPC2 in $cm^{-3}$ for the size between 12 nm and 1 $\mu$m ($N_{>12}$) on ALADINA in Ny-Ålesund between 24 April and 25 May 2018. The colour bar ranges from $0\,cm^{-3}$ (blue) to $4000\,cm^{-3}$ (red).



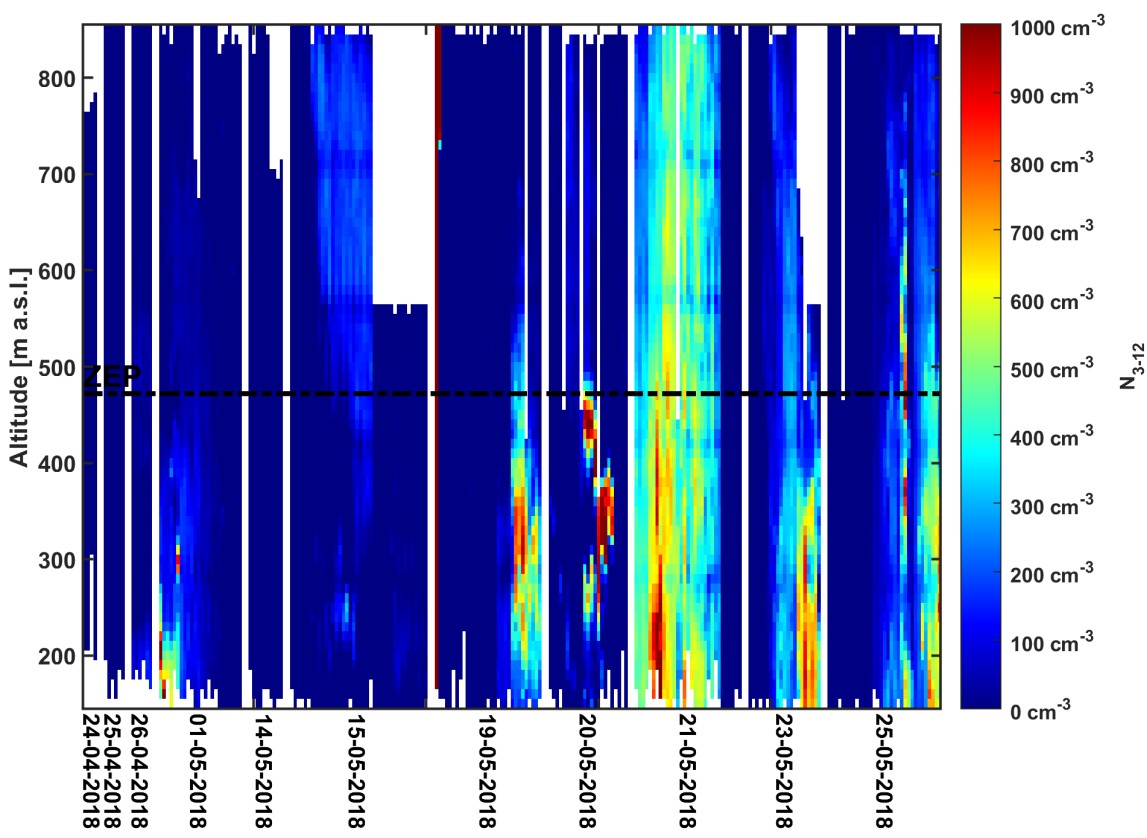

**Figure A3.** The same as Fig. A1, but time series of vertical profiles of aerosol particle number concentration calculated from the difference between CPC1 and CPC2 in $cm^{-3}$ for the size between 3 nm and 12 nm ($N_{3-12}$) on ALADINA in Ny-Ålesund between 24 April and 25 May 2018. The colour bar ranges from $0\,cm^{-3}$ (blue) to $1000\,cm^{-3}$ (red).



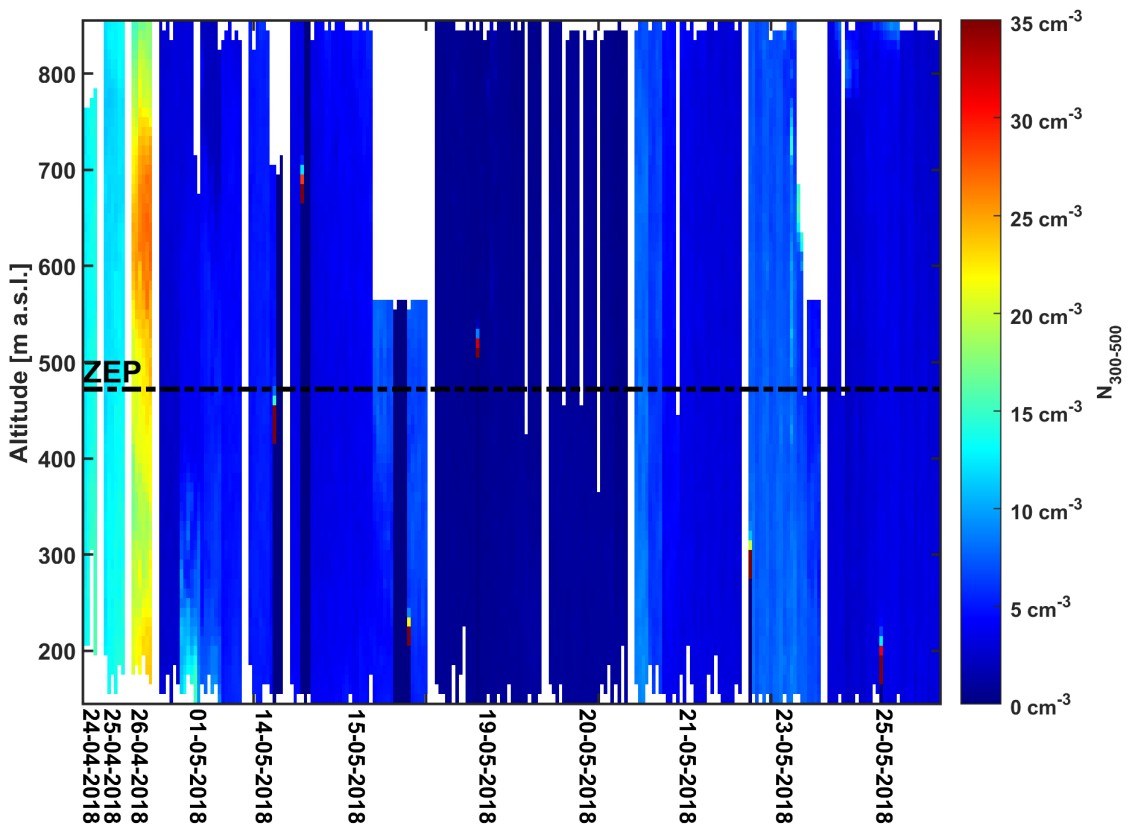

**Figure A4.** The same as Fig. A1, but here vertical profiles of aerosol particle number concentration measured with the first channel of the OPC within the size of 300 and 500 nm ($N_{300-500}$) in cm$^{-3}$ on ALADINA during the field period in Ny-Ålesund between 24 April and 25 May 2018. The colour bar ranges from 0 cm$^{-3}$ (blue) to 35 cm$^{-3}$ (red). The other channels of the OPC are not considered for the study shown here, as the number concentrations larger than 500 nm were out of the detection limit.





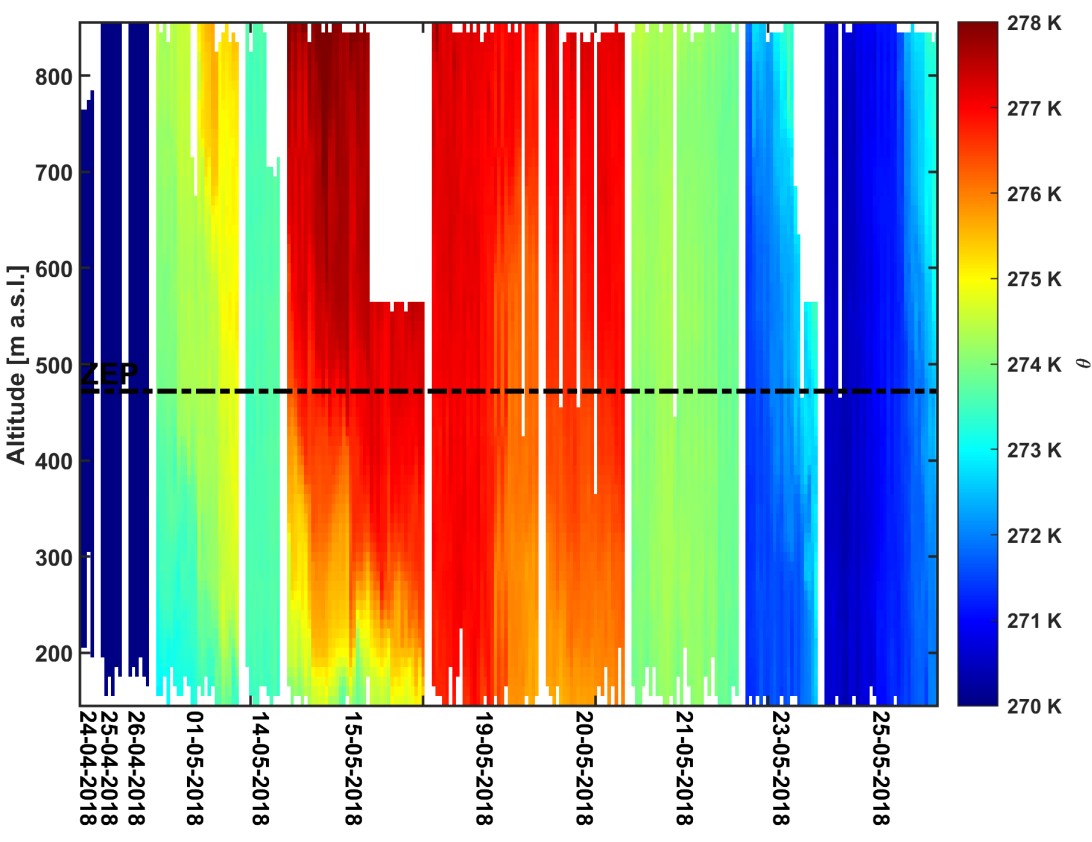

**Figure A5.** The same as Fig. A1, but here vertical profiles of potential temperature $\theta$ in K measured with ALADINA in Ny-Ålesund between 24 April and 25 May 2018. The colour bar ranges from 270 K (blue) to 278 $\mathrm{K}^{-3}$ (red).

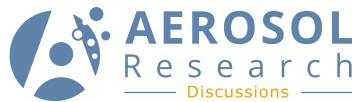

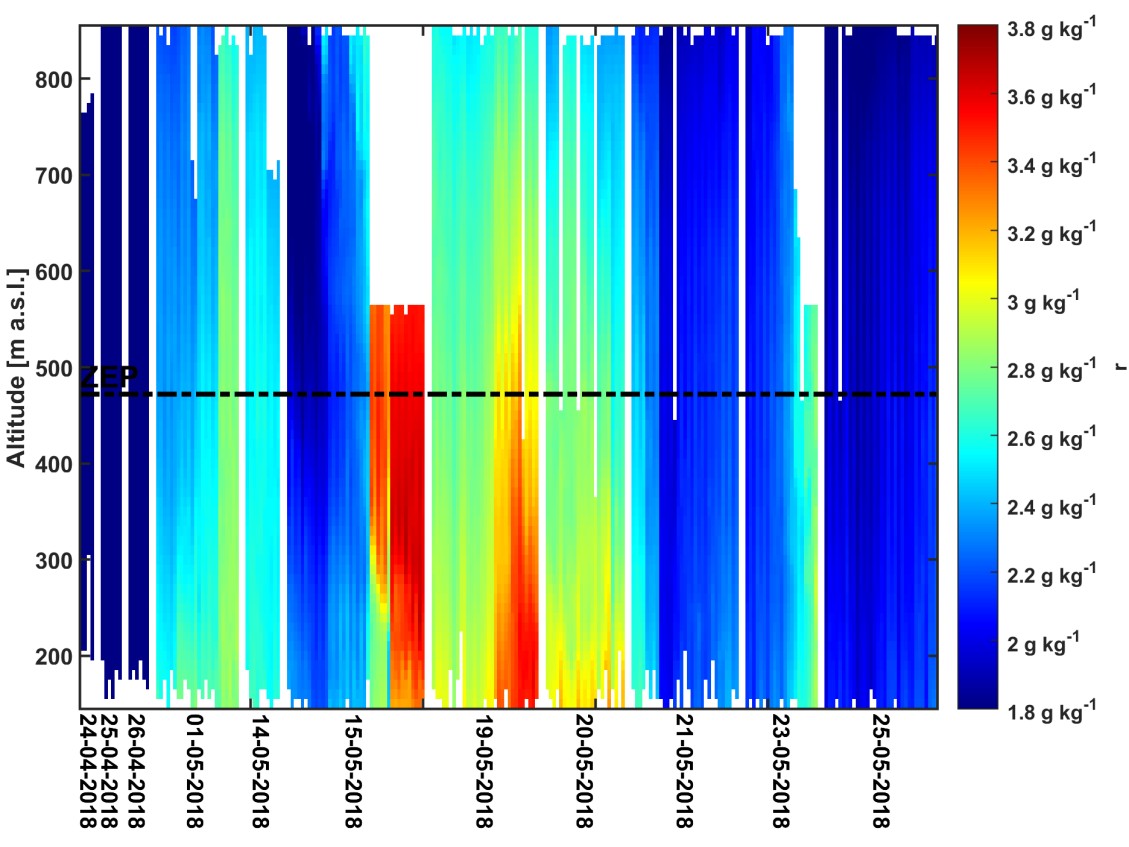

**Figure A6.** The same as Fig. A1 but now valid for vertical profiles of water vapour mixing ratio $r$ in $g\,kg^{-1}$ measured with ALADINA in Ny-Ålesund between 24 April and 25 May 2018. The colour bar is between $1.8\,g\,kg^{-1}$ (blue) and $3.8\,g\,kg^{-1}$ (red).

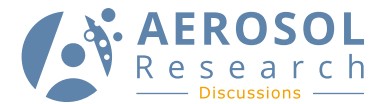

*Data availability.*    The UAS and AWIPEV data are publicly accessible at the PANGAEA data publisher: ALADINA in Harm-Altstädter et al. (2022), MASC-3 in Schön et al. (2022b) and ground based meteorology observations in Maturilli (2018a–2018d). SMPS data from GRU are available under request to Mauro Mazzola or Rita Traversi. Contact Radovan Krejci for MAAP and DMPS data from ZEP and Kihong Park for nano-SMPS data from ZEP.

*Author contributions.*    AL, BW and JB initiated the project. LB, RK, FP, KB and AP prepared ALADINA for the polar field activity, including
calibration of aerosol sensors, setting up the new design and sensor development. KB, FP and LB were responsible for planning the flight strategy, data acquisition and post-processing of ALADINA. BHA, KB, MS, BW, JB, LB, RK and AP participated in the field campaign and collected data. BHA, AL and BW analyzed and interpreted the data and contributed to text. KB, BHA, and MS contributed to the figures. RKR provided DMPS and MAAP data, MM and RT handed over SMPS data from Gruvebadet and KP sent nano-SMPS data. BHA wrote the main text and all authors reviewed the manuscript.

*Competing interests.*    The authors declare that they have no conflict of interest.

*Acknowledgements.*    This work was funded by the German Research Foundation (DFG) under the project number LA 2907/5-3, WI 1449/22-3, BA 1988/14-3. We thank Markus Hermmann from TROPOS for his help in setting up the aerosol instrumentation in ALADINA. The authors gratefully acknowledge Roland Neuber and Christoph Ritter from Alfred Wegener Institute (AWI) for their support during preparation of the field campaign. We thank the AWIPEV Base and crew for hosting the participants and in particular Piotr Kupiszewski and Rudolf
Denkmann for valuable assistance at site. A special thank to Rune Storvold from the Norwegian Research Centre (NORCE) for enabling access to the facility at the airport. The data were analysed in cooperation with the transregio project TRR 172 (AC)3, funded by the German Research Foundation under project ID 268020496. The aerosol research at Zeppelin was supported by a National Research Foundation of Korea Grant from the Korean Government (MSIT; the Ministry of Science and ICT) (NRF-2021M1A5A1065425) (KOPRI-PN23011). The research activity at Gruvebadet was made possible by Projects PRIN- 20092C7KRC001 and RIS 3693 "Gruvebadet Atmospheric Laboratory
Project (GRUVELAB)" and by the coordination of National Council of Research (CNR), which manages the Italian Arctic Station "Dirigibile Italia" through the Institute of Polar Sciences (ISP).



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
