# Peer review of "Spatial distribution and variability of boundary layer aerosol particles observed in Ny-Ålesund during late spring in 2018"

_Aerosol Research, 2023_

## Author Response (AR1)

**First of all, the authors acknowledge both referees for their valuable contributions that will improve the outcome of our study. This means in particular improvements by shortening the text in order to concentrate on the main objectives. The authors agree that the described methods are long which is given by the fact of a high complexity of different data sets that contribute to different disciplines- ranging from meteorological parameters and aerosols of different sizes, measured at different locations and altitudes as well as focusing on the results that are based on the performance of two unmanned aerial systems at a complex Arctic coastal site.**

**In the following, the _author's response_ is directly given with respect to the individual referees' comments, starting with reviewer 1 and reviewer 2 as well as additional changes that were made by the authors in consequence of minor corrections of the previously submitted manuscript or adaptions that were conducted in result of critical comments of the reviewers. Please find all changes in the marked-up manuscript, enclosed to this response.**

**Reviewer 1**

1. "The paper by Harm-Altstädter et al. addresses an important topic of small-scale aerosol distribution and dynamics affected by different atmospheric boundary layer (ABL) properties. The study, which focused on the transport and mixing of ultrafine aerosol particles (UFPs) as possible sources of new particle formation (NPFs), was performed by combining measurements of unmanned aerial systems (UAS) with continuous observations from the ground at different altitudes, at the observatories of Gruvebadet (67 m asl) and Mount Zeppelin (472 m asl). I believe that the topic is interesting and there are elements of innovation especially because the aerosol distribution is not frequently studied with these approaches. The paper is written and developed very well and should be suitable for publication after providing a better context and clarifying few details.

General comments:

The manuscript is unnecessarily long in some parts (1 Introduction, and 2 Description of the measurement site, instrumentation, and data availability) and various pieces of information are repeated several times throughout the text. These risks making the manuscript difficult to read and boring in parts."

_Thank you very much for your positive feedback! We agree that the text of the first two parts of the manuscript regarding introduction and description of the methods is long and detailed. However, as the UAS measurements have different approaches like assessing atmospheric boundary layer properties and aerosol particles with different sizes, the introduction takes into account all considered aspects, including the state of the art of mobile measurements performed in the Arctic. We have shortened the text and checked for unnecessary repetitions._

2. "I recommend the authors to reduce sections 2.3.1 and combine sections 2.3.1 and 2.3.2 in section 2.3."

_The authors thank the reviewer for the suggestion to merging the two sections into one in order to reduce the total length of section 2.3 of the manuscript and by focusing on the main parts. The authors decided to briefly refer to other studies that have introduced the setup of ALADINA in more detail that was newly adapted for harsh polar environmental conditions. The main reduction was done by removing information about flight procedure with ALADINA and by solely mentioning the meteorological sensors which are of minor importance in this study. However, as a profound introduction of the aerosol compartment is of vital importance for understanding the results presented in the manuscript, the authors decided to maintain the detailed descriptions of the aerosol sensors regarding the setup, accuracy and performance of the two CPCs, OPC and AE51 (which was not used for the analysis due to low background concentration that resulted in a poor performance of the detection limit of the attenuation)._

_Subsection 2.3.1 UAS ALADINA and Subsection 2.3.2 UAS MASC-3 were removed as individual titled subsections and the whole content was shifted to Sect. 2.3_

*The following sentences were removed or changed accordingly:*

*removed: p. 8, l. 180-183*

The aircraft is insulated, and batteries are stored in a side board, which guarantees a fast turnaround time of around 20 min, as batteries are easily replaceable in the field and data can be quickly saved via downlink after each landing. The inner compartment is heated to a stable temperature range in order to assure the reliability of the instruments' working flow, which is especially important for the aerosol sensors.

*The two sentences hereafter written in the old manuscript, p. 8, l. 185-188*

Different types of temperature sensors and humidity sensors, as well as a multi-hole probe are installed for the calculation of air temperature, humidity and the 3-D wind vector, and the sensors are mounted at the tip of the aircraft in order to assure undisturbed measurements of the air probe. In addition, two pyranometers are integrated into the UAS, one on top and the other one underneath on the fuselage, which enable measurements of the incoming solar radiation and reflex radiation, respectively.

*were rephrased to:*

Different types of temperature sensors, humidity sensors, as well as a multi-hole probe and two pyranometers are installed for the calculation of air temperature, humidity, 3-D wind vector and radiation properties. More information about the meteorological measurement unit is available in Bärfuss et al. (2018).

*The following sentences were deleted p. 8, l. 189-191*

A detailed description of the meteorological measurement unit is not in the scope of the current study, as information are provided in Barfuss et al. (2018) and Lampert et al. (2020). The performance of the newly designed aerosol instrumentation environment is also shown in Lampert et al. (2020).

3. The captions are very long, can't you take some information to the text? This applies for all long figure captions (Figure 1; Figure 2; Figure 3; Figure 5; Figure 7; Figure 9; Figure 10). The same for Table1.

*We reduced the length of the image captions (see below). However, we still keep the detailed lists of*
*abbreviations and specific units in the captions, instead of providing them in the main text. We believe that this helps maintaining the readability of the main text.*

*The contents of the mentioned captions were reduced:*

*Caption Fig. 1: Here only the main stations are considered in the text now*

*old version: p. 5, l. 1-7*

The map represents the topography (grey shading in colour bar) of the investigation area around Ny-Ålesund that belongs to the Svalbard Islands, Norway. Research flights were performed with the two UAS ALADINA and MASC-3 at the local airport (red point, appr. 40 m a.s.l.) in April–May 2018. The main flight patterns were performed in parallel to the airport and crossing the coast via horizontal legs (black lines). Aerosol in-situ data are used from Gruvebadet (67ma.s.l.) and Zeppelin Observatory (472 m a.s.l.) and meteorological data is taken into account from the AWIPEV station, within the research village that is indicated by the yellow circle. Future planned studies will consider turbulent flux measurements at Old Pier as well. The main wind directions are represented by the white arrows: one from the Kongsfjord glacier (SE) and a temporary wind direction coming from the glaciers, south of the village (SSW).

*new version:*

The map represents the topography (grey shading in colour bar) of the investigation area around Ny-Ålesund (yellow circle). Research flights were performed with the two UAS in parallel to the airport (red point, appr. 40 m a.s.l.) and crossing the coast via horizontal legs (black lines) in April–May 2018. Aerosol in-situ data are used from Gruvebadet (67 m a.s.l.) and Zeppelin Observatory (472 m a.s.l.)

and meteorological data is taken into account from the AWIPEV station.

*Caption Fig. 2: Here the detailed descriptions of the smaller pictures were reduced.*

*old version: p. 6, l.1-7*

For a better orientation and information on snow conditions, selected pictures are shown of the investigation area which were taken during the field period. (a) The two research stations Gruvebadet and Zeppelin Observatory from a bird's eye view, taken at the end of the campaign on 25 May 2018. The UAS MASC-3 was used for studying turbulent properties (b, left hand-side) and ALADINA was operated for investigating of aerosol particles linked to atmospheric boundary layer properties (b, right hand-side). At the beginning of the campaign, the site was covered with snow but the snow melt occurred and the water area around the coast was completely ice free (c). Pictures: © TU

Braunschweig

*new version:*

(a) A bird's eye view of the two research stations Gruvebadet and Zeppelin Observatory. (b) The two UAS MASC-3 (left hand-side) and ALADINA (right hand-side) during research flights. (c) During the field experiment, the snow melt occurred and the water area around the coast was completely ice free.

Pictures: ©TU Braunschweig

*Caption Fig. 3: As the data displayed in Figure 3 consists of a large variety of measured parameters that are obtained from different stations as well as distinguish systems (e.g. AWIPEV, Zeppelin, two UAS), a drastic shortening might cause misunderstandings or at least might increase the complexity in*

*capturing and interpretation the data. For this purpose, the authors intended to keep a detailed indication of the specific legends in the text of the caption. Thus, only a small reduction in terms of summing up the wind properties and CBL measured at AWIPEV and different size bins measured with the nano-SMPS at ZEP were applied in this case.*

*old version: p. 11., l. 1-10*

Time series of selected parameters that contribute to the UAS study in Ny-Ålesund, valid for the period between 00:00 UTC on 24 April and 00:00 UTC on 2 May 2018. From top to bottom: wind speed $FF_2$ in $ms^{-1}$ calculated in a 2 h average, measured at the AWIPEV station at 2 m level (Maturilli, 2018a, b). The wind arrows represent the wind direction in the same time range, respectively. Cloud base height (CBH) in m for 10 min interval (Maturilli, 2018c, d) derived from the AWIPEV station (black dot) in comparison with the sequences of ALADINA profiles (cyan dot) and levels of horizontal legs based on MASC-3 (magenta triangle). Equivalent black carbon mass concentration (eBC) is estimated from a MAAP (Multi-Aerosol Absorption Photometer) at the Zeppelin Observatory in 1 min (green dot) and averaged for 1 h (black line). Aerosol particle number concentration (N) in the size of 3 to 12 nm ($N_{3-12}$, red dot), 12-25 nm ($N_{12-25}$, green dashed line) and 25-50 nm ($N_{25-50}$, blue dashed line) measured with a nano-SMPS in 3 min intervals at the Zeppelin Observatory. The blue shading represents the three measurement days of ALADINA that are considered for a deeper analyses in the first case study (Case I, Sect. 3.3).

*new version:*

Time series of selected parameters valid for the period between 00:00 UTC on 24 April and 00:00 UTC

on 2 May 2018. From top to bottom: wind speed $FF_2$ in $ms^{-1}$ and wind direction in a 2 h average at 2 m level, as well as cloud base height (CBH) in m for 10 min interval (black dot), all derived from the AWIPEV station and here shown in comparison with periods of ALADINA flights (cyan dot) and MASC-3 flights (magenta triangle). Equivalent black carbon mass concentration (eBC) is estimated from a MAAP (Multi-Aerosol Absorption Photometer) in 1 min (green dot) and averaged for 1 h (black line), and aerosol particle number concentration (N) was derived for different sizes from a nano-SMPS in 3 min intervals, both measured at the Zeppelin Observatory. The blue shading represents the three measurement days of ALADINA that are considered for a deeper analyses in the first case study (Case I, Sect. 3.3).

*Caption Fig. 5: We agree that the text of the caption is very detailed so that we have removed some parts into the running text of the manuscript. In consequence of questioning this kind of illustrations by the two referees, we think that more description is necessary at this point. Most likely the term "spatial distribution of aerosol particles" at the very beginning of the text might have caused misunderstandings or was not detailed enough in reproducing the results of the illustrations.*

*For this purpose, here are some comments for pointing out the idea of the presentation style that was used for the aerosol data in this specific way.*

*The main intention  -as well as one of the main goals of the study- is to reproduce the measured total aerosol particle number concentration at both fixed sites (GRU and ZEP) in comparison with the vertical profiles that are derived from the ALADINA aerosol compartment. As the possibility to produce*

*such a 4-D picture of aerosols in the spatial scale relies on methods of airborne profiling, the authors intended to maintain the picture in this special manner as it shows the potential for filling missing gaps that occur in consequence of observations at different altitudes that are strongly affected by meteorological properties.  As both reviewers address to change the pictures or at least mentioned that "Figure 5 and Figure 6 are very complex and difficult to follow" , we included additional information*

*in the text that should help to better follow the results displayed in the picture(s). Please take into account the changes made by the authors at the end of the author's response, here marked up in orange on p. 13 f.*

*Nevertheless, the text of the caption was changed as following:*

*old version: p. 14, l. 1-11*

[revised manuscript text omitted]

*The following sentences were replaced to the main text in order to maintain the basis information that is essential in order to understand the presented results.*

[…] More precisely, the histograms are based on vertical profiles of aerosol particle number
concentration in different sizes $N_{3-12}$, $N_{>12}$, $N_{300-500}$, potential temperature θ and water vapour mixing ratio r between a typical height of 150–850 m a.s.l. that are further presented in Figs. A1–A6. This chosen altitude area excludes surface measurements with ALADINA and due to safety reasons, the majority of the profiles started at an altitude of 100 m above ground level (a.g.l.) and as the airport is located at a level of around 40 m a.s.l., all profiles are bordered in the specific altitude above 150 m
a.s.l. in order to provide the highest statistical relevance. Further, the black dashed line indicates the height of the Zeppelin Observatory. Note that the maxima of $N_{3-12}$ and $N_{300-500}$ are not provided in the graph in order to provide a better readability of the analysis, as they are far outside of the measurement range.

*Caption Fig. 9: The main reduction was carried out by removing the time series and specific colour codes of the selected profiles in Figure 9. In order to maintain a clear classification for the readers, the nominations were shifted into the main text.*

*old version: p. 21, l. 1-9*

[revised manuscript text omitted]

*In order to help the reader capturing the specific altitude levels that are presented in the figure, the text was expanded by the following:*

Figure 10 shows the horizontal distribution of $N_{3-12}$ along legs that were performed at three different constant altitudes (marked in white and from left to right: 173–192 m a.s.l., 314–334 m a.s.l., 458–478 m a.s.l.) during four measurement flights (from top to bottom) with ALADINA between a period from 11:44 to 14:34 UTC on 20 May 2018. Each flight pattern (black line) consists of legs that cross the coast in direction from the airport to the sea, a full operation above open water by heading to SE with a distance of around 2 km. The turnaround from the sea back to the airfield is used for achieving the next altitude level, thus this part is not considered for the study as the variability may be attributed to changes in the altitude.

*Caption Tab.1: The text written in the caption Table 1 was significantly reduced by focusing on the main goal of the study, consisting now of the following lines:*

*old version: p. 10, l. 1-8*

This table summarizes the data availability of the study presented here. The UAS ALADINA performed 49 research flights on eleven different days, and a total sample of 230 vertical profiles was enabled during the polar field campaign in Ny-Ålesund between 24 April and 25 May 2018. The second UAS MASC-3 was operated during six common measurement days with ALADINA, and the study considers vertical profiles of the calculated horizontal wind. Ground based aerosol data is continuously available from SMPS at the Gruvebadet Observatory (GRU). Aerosol data measured with DMPS and nano-SMPS are not complete on a daily basis at the Zeppelin Observatory (ZEP) during the period. Equivalent black carbon mass concentrations (eBC) is calculated from MAAP at the Zeppelin Observatory. "NO" means not operated, "NA" stands for not available, "X" represents data availability during the individual ALADINA measurement flights and the data are freely accessible in Harm-Altstädter et al. (2022).

*New version:*

As one of the main objectives of the study is on filling missing information about the spatial distribution of aerosols between the two fixed long-term observatories GRU and ZEP, this table shows the data
availability of additional instrumentation that was deployed during the ALADINA period. "NO" means not operated, "NA" stands for not available, "X" represents data availability of the instrumentation during the specific days when research flights were performed with ALADINA.

4. "Specific comments:

• L153: Further?

     • L153-155: would move this sentence after L162.

     • L156: where eBC mass concentration were collected?"

*Thank you very much for giving us the hints! Due to providing answers by addressing similar points, we have summed up the answers into one part. Nevertheless, here are short comments directly linked*
*to the points. First bullet point: Yes indeed, "Further" is not correct at the beginning of the sentence written in the previous version of the manuscript. This resulted in a previous rearrangement made by the authors that led to a wrong order of the sentences that are pointing out by you as well in bullet point 2.*

*After shifting those sentences in a correct order, the third point should be answered as well, as the*
*eBC data was calculated from the MAAP that is deployed at the Zeppelin observatory, which was not clearly stated in the previous form of the manuscript.*

*The text was changed to:*

A scanning mobility particle sizer (SMPS, model 3034, TSI Inc., USA) is deployed at GRU, which measures in the particle size range between 10 and 470 nm (Hogrefe et al., 2006; Lupi et al., 2016).

At ZEP, the aerosol size distribution is derived from a combination of differential mobility particle sizers (DMPS) in the size of 5–810 nm and 10–790 nm. Further, UFP of different sizes are determined with a nano-SMPS (nano-scanning mobility particle sizer) at ZEP, which is a combination of a nano-DMA (differential mobility analyzer, model 3085, TSI Inc., USA) and a CPC (condensation particle counter, model 3776, TSI Inc., USA) in 3 min temporal intervals. In order to provide information about possible
local pollution at the investigation site, eBC mass concentration data are used, which are calculated from the aerosol light absorption coefficient measured with a multi-angle absorption photometer (MAAP, model 5012, Thermo Fisher Scientific Inc., USA), also deployed at ZEP. The Zeppelin Observatory in its full facility was recently presented in Platt et al. (2022) and shows more information about the instrumentation available at site.

5. "L221: what kind of meteorological measurements? Please, specify the parameters."

*The meteorological parameters derived from MASC-3 are similar to the ALADINA observations of air temperature, humidity and 3-D wind vector, as it is explained in the text in the following lines. In order to prevent unclear descriptions, the sentences are sorted in a new order so that the information about*
*the specific measurements comes at first.*

*For this purpose, the following sentences were rephrased: p. 9, l. 220-224*

The UAS MASC-3 (Multi-Purpose Airborne Sensor Carrier) in its third version (Fig. 2b) was developed by the Eberhard Karls Universität Tübingen and is equipped with a sensor system for meteorological measurements and described in more detail in Rautenberg et al. (2019). MASC-3 has a wingspan of 4
m, a weight of 5–8 kg, depending on the payload, and a maximum flight duration of 2 h. The sensor system consists of a multi-hole probe, a fine-wire platinum resistance thermometer and a slower digital humidity sensor.

*new version:*

The UAS MASC-3 (Multi-Purpose Airborne Sensor Carrier) in its third version (Fig. 2b) was developed
by Tübingen University (Germany). It has a wingspan of 4 m, a weight of 6.5 kg, a maximum flight duration of 2 h, and is described in more detail in Rautenberg et al. (2019). MASC-3 is equipped with a sensor system that consists of a multi-hole probe, a fine-wire platinum resistance thermometer and a slower digital humidity sensor. The high resolution 3-D wind vector and air temperature can resolve turbulent fluctuations.

6. "L243: Fig. 3b and Fig. 4b (and throughout section 2.4)? I only see Fig. 3 and Fig. 4"

"Figure 3 and Figure 4 show different time scale, daily and every two days. Please uniform them."

*The authors have merged the two comments into one as they relate to the same figures Fig.3 and Fig. 4. The authors have done the following changes: the individual boxes of the figures are now titled from*

*"a" to "d" (see Fig. 3 and Fig. 4 in the marked-up manuscript). This was done in order to better identify the parameters in the text. Further, the time series are changed into a daily basis at the x- axis in order to obtain a uniform illustration. For this purpose and in order to ensure a good readability by not reducing the size of the labels, the x-label is rotated from initially 45° to 90°.*

7. "Figure 5 and Figure 6 are very complex and difficult to follow."

*The authors thank the reviewer for pointing out the difficulties in understanding the results displayed in the two illustrations Fig. 5 and Fig. 6.*

*Please read the comments above addressing the caption of Fig. 5 (p. 4 first two paragraphs) as well as the response to reviewer 2 at p. 11.*

8. "Section 2.4 is a bit confusing; it is not clear which days and which types of measures are available. Considering the short duration of the measurement campaign, it would perhaps be clearer to modify Table 1 by providing an overview of all days and data available (not only of US ALADINA)."

*We have changed the text in the caption of Table 1 to better explain the aim of the study. The main*

*goal is to link the aerosol properties at the vertical scale in order to illustrate possible effects of atmospheric boundary layer processes on the vertical mixing between the two fixed sites (Gruvebadet and Zeppelin). These sites provide long-term continuous data and are located at different altitudes. The study of the vertical aerosol distribution is the main objective of this paper. MASC-3 is used for additional turbulent studies. This results in a different flight mission, which implies long horizontal legs*

*instead of profiling. For this reason, it was chosen to highlight the large number of vertical profiles that have been performed with ALADINA and to compare the data availability of ground data with respect to the specific ALADINA measurement days.*

9. "L263-265: I'd move this sentence to the beginning of the section 2.4, where the authors provide the general overview (Table 1)."

*The authors want to thank for this suggestion of moving the sentences to the beginning of the section. However, the first part considers the flight missions in a general manner that were performed in a different way by the two UAS. The paragraph ranging from l. 255-265 takes into account explanations for the ALADINA data availability in consequence of unfavourable weather conditions that impact the*

*feasibility of the research flights.*

*We made following change:*

[…]

During the periods from 27–30 April 2018 due to heavy snowfall (Fig. 3b) and in the presence of low level clouds and high wind speed (Fig. 4b), no field activity was carried out on 16–18 May 2018 and on

24 May 2018.

*The following text was shifted to the first paragraph of Sect 2.1:*

[…] 2018 in a transition period between spring and early summer, thus influenced by snow melt, which can be further seen in the reduced snow covered surfaces (Fig. 2). However, from 2 May 2018 to 13

May 2018, no measurements were performed due to technical reasons. For safety reasons […]

10. "L324: period and not colon."

*Probably we misunderstood the suggestion to replace period by colon. But we have done following change to end this sentence.*

*old version: p. 16, l. 324*

*[…] can be explained by the following**:** The first three*

*new version*

*[…] can be explained by the following**.** The first three*

11. L 536: the authors write that the frequency of occurrence of NPF events during the ALADINA period was 22% while in the abstract, L8-9, they write "A high frequency of occurrence of NPF was observed, namely on 55 % of the airborne measurement days". Please explain.

*The text in the conclusion relates to the findings of the study of Lee et al. (2020) who calculated the*
*occurrence of frequency for NPF with 23% for the overall period between October 2016 and December 2018. Further, it is mentioned in l. 532 that this is similar "during the ALADINA period in case of considering only NPF events of class I" which is not a feasible practice, as sporadic appearances are not taken into account in the case if not matching a typical "banana shape". We pointed out in the first part of the results in Sect. 3.1 that a high difference occurs in consequence of*
*considering different NPF classes.*

*We have moved the whole paragraph to the end of Sect. 3.1 and have changed the expression "well known" with "the" (see remark in the text below):*

*old version p. 16, l. 314-322 now at the end of Sect. 3.1:*

"Summarizing the observations during the presented 22 measurement days in Figs. 5–6, UFP occurred frequently on 55% of the 12 different measurement days, but the appearances of UFP are mainly linked to non-defined NPF events, thus might not have been assessed after the typical classification for NPF events. Only three NPF events may have been classified as NPF event with subsequent growth rate which further results in a so called "banana-shape" (Heintzenberg et al.,
2007). However, for most of the events, the particles' growth was interrupted and lasted until around midday of the following day, for instance during the observations on 30 April-1 May 2018, 14-15 May 2018 as well as on 21–22 May 2018. By considering only  the classic NPF event days, the frequency of occurrence is significantly reduced to a value of 23%, as the classification is only applicable for five measurement days, which, however, coincides with the study of Lee et al. (2020)
who considered a two year data set."

*The authors see the need to change this in the abstract in order to avoid any mixing of the two different aspects by considering typical NPF events with subsequent growth rates, that are contributing to a class I event day in comparison with high occurrence of UFP that appear on a short period of time*
*that are linked to NPF events of class II or non-defined events. In our mind, this should be easily solved by using UFP instead of NPF in the abstract, as a more detailed description is following in the discussion of Sect. 3.1.*

*Old version:*

A high frequency of occurrence of **NPF** was observed, namely on 55% of the airborne measurement
days.

*To the new version:*

A high frequency of occurrence of **UFP** was observed, namely on 55% of the airborne measurement days.

12. L 534-544: I recommend moving this sentence somewhere in Discussion. Also, where is it mentioned about GR in the paper?

*Thanks for this suggestion. The comparison with the study of Lee et al. (2020) is of high relevance for the ALADINA measurements, as it covers a long period of UFP observations at ZEP which further*

*estimated particle growth rates based on the nano-SMPS data. The GR and its high variability from 0.48 to 6.54 nm h$^{-1}$ was cited in order to show that at some occasions a high nucleation rate with subsequent and fast growth of the particles is obviously possible at polar sites.*

*The whole part was removed from the conclusion, p. 26, l. 536- p. 27, l. 544*

The UFP occurrence frequency was estimated to 23 %, which is similar to studies at continental sites and was matching the frequency of occurrence during the ALADINA period in case of considering only NPF events of class I. Further, the calculated mean growth rate (GR) for the particle size from 3 to 25 nm was low with 2.66 nm h$^{-1}$,  thus significantly lower in comparison with other sites in the world (e.g., Nieminen et al., 2018).

Interestingly, the study specified high variances of GR for the size of 3–25 nm with 0.48 to 6.54 nm h$^{-1}$, thus UFP may grow on a rapid pace during some occasions, which is generally not assumed for polar studies. The measured highest values of the GR are similar to rural observations and those high GR were temporarily measured during the ALADINA period as well

*The text in the new manuscript is in a shortened version without mentioning of nucleation rate and ABL and was shifted to the last paragraph of Sect. 3.1:*

[…] By considering only well known classic NPF event days, the frequency of occurrence is significantly reduced to a value of 23%, as the classification is only applicable for five measurement days, which, however, coincides with the study of Lee et al. (2020) who considered a two year data set. The study calculated a mean growth rate (GR) of 2.66 nm h$^{-1}$ for the particle size of 3 to 25 nm that is significantly lower in comparison with other sites in the world (e.g., Nieminen et al., 2018). Interestingly, the authors indicated high variances of the measured GR ranging from 0.48 to 6.54 nm h$^{-1}$, thus UFP may grow on a rapid pace during some occasions, which is generally not assumed for polar studies. The measured highest values of the GR are similar to rural observations and those high GR were temporarily measured during the ALADINA period as well.

**Reviewer 2**

"The current paper is outstanding on the quality of the science and the data collection and interpretation. However, I am afraid the presentation is very poor. In a paper, it is important not only to collect the data - but also to present them well."

*The authors acknowledge the reviewer for the positive judgment.*

"I suggest two main improvment that are really necessary for this paper to be published. Failing to address this, this paper cannot be accepted.

1) The paper is too diluted, the reading is heavy and at the end one cannot find the major information. Please consider some bullet points at the end of the discussion with 5 major sentences stressing the major findings"

*We included a short summary with the main findings as suggested at the end of Sect. 3.1:*

The main findings of the performed UAS field experiment are briefly summarized before finally concluding the study in Sect. 4:

- The study presents a unique data set of aerosol particles and meteorological parameters in the spatial scale, measured with the two UAS ALADINA and MASC-3 that are linked to long-term
measurements of aerosol particles observed at two different altitudes.

- The integrated setup of ALADINA allows to investigate different sizes of aerosols, ranging from UFP to accumulation mode, thus provides a high potential of covering the spatial distribution of different phenomena like sources of NPF, mixing and transport of UFP, as well as distribution of larger particles that may have been transported to the site via long-range transport, for instance within the Arctic haze
period.

- Within the UAS period, UFP occurred frequently in Ny-Ålesund but mainly on a short period of time, and these days would not have been identified as NPF events if surface measurements were taken into account alone. By considering the summary of all performed vertical profiles of UFP, highest number concentrations appeared near ground and were strongly affected by a stably stratified ABL. In
cases, when UFP were observed at both research stations, accumulation mode particles played only a minor role in the aerosol population, thus leading to the assumption that during the start of the UAS period, when the Arctic haze was in the last phase, the large pre-population of accumulation mode particles inhibited the particles' growth.

-  By reflecting the measured potential temperature and mixing ratio in the vertical scale, ABL
properties play a crucial role on the vertical distribution of aerosols, so that the observations at Gruvebadet differ in many cases from the measurements at Zeppelin.

- Other case studies show that UFP can coexist at different altitudes in consequence of a stably stratified ABL which was further supported by investigations of a high variability of UFP in the horizontal scale.

"2) Figures. The figures need all redrawing, they are not clear and some mirror images are impossible to look at. figure 5 and 6 are not pubblishable, please consider putting it all in 2d normal smps plotting with site a and site b below and above, with the vertical profiles in the middle, and simply state on the left chart what is what. (like figure A1 and A2 that are clear and simple)"

*The new aspect of our research is that we have available data not only as time series, but also the vertical dimension. We agree that the representation is unusual but we acknowledge your feedback and have improved the figure captions to better explain our plots. We hope that the reader gets an understanding of the particular strength of ALADINA observations. Representing vertical profiles as*
*time series is possible, of course, but are not directly comparable to the measurements at the fixed locations. Representing the ALADINA measurements as vertical profiles is also possible, but this*

*would not allow providing information on the high temporal aerosol variability simultaneously with the observations at the fixed locations. The authors decided to provide time series of the vertical profiles of the specific parameters which were measured with ALADINA solely in the appendix Fig. A1-A6, as the main focus in on linking observations at the different research stations filled with missing information about the vertical distribution between both sites. On the other hand, we would miss out important information about growth processes that have already started before the flight period, in case of only providing vertical profiles with ALADINA and excluding ground-based measurements in the direct comparison.*

*In order to increase the understanding of the figures, the three different data sets were indicated via grey boxes at the position in the graphs and following lines were added to the text at the beginning of Sect. 3.1.:*

In order to discuss the spatial distribution of aerosol particles at the complex site, the time series of aerosol particles are shown in a 3-D representation in Figs. 5–6. Ground based data was derived at two different altitudes: first close to the surface from a SMPS at GRU (Figs. 5–6a) and secondly from a DMPS at ZEP (Figs. 5–6b), measured at Mount Zeppelin. The continuous data is further compared with vertical profiles of $N_{3-12}$ that are displayed in the background (Figs. 5–6c) for a potential link between the two research stations. The figures are further separated into two main episodes, matching the same time slots as presented in Sect. 2.4, for the first part between 24 April 2018 and 2 May 2018 (Fig. 5), and for the second part considering observations from 14 May 2018 until 26 May 2018 (Fig. 6).

"figure 11 and figure 12 are interesting, what is the brown in the sea, river influnce or ice? if ice put it in white or blue, it may help the reader to see what you are doing."

*Probably the reviewer is referring to Fig. 10 and Fig. 11. The underlying picture is a satellite image (source Earthstar Geographics as an output from a used matlab tool), which is mentioned at the bottom of each graphs. We agree that the source of the map is barely readable so that we included the information in the caption of the two figures. The sea was completely ice free during the end of the period (see pic. Fig. 2c), but it can not be ruled out whether there existed remains of ice during the time when the satellite image was taken or not. The brown colour in the sea is most likely sediment transported into the fjord.*

*We added the information about the source of the map into the caption of Fig. 10 and Fig. 11:*

Source of the satellite image: Earthstar Geographics

Additional changes made by the authors:

*Here the authors want to show additional changes that were done due to improve the readability or in consequence of parts that are not relevant for the study.*

Additional changes made by the authors at the end of the Introduction:

[…] and a study of **eBC measured** during a day affected by a higher degree of local pollution on 23 May 2018 (Case III). p. 4, l. 116

*As the case study was changed to only showing UFP, the sentence has to be rephrased*

[…] and a study of **increased UFP that appeared** during a day affected by pollution on 23 May 2018 (Case III).

*Additional changes made by the authors for caption Fig. 4:*

*As the authors decided not to use the AE51 data in the manuscript, the last sentence has to be*
*rephrased. p. 12, l. 3-5*

old: […] that considers

*New version:*

[…] that considers the occurrence of $N_{3-12}$ along with a pollution event.

*Additional changes made by the authors for caption Fig. 6:*

*The caption was reduced to the main comments, as the full description of the graph is given in the caption of Fig. 5 as well as in the main text.*

*Old version: p. 15, l., 1-4*

The same parameters as shown in Fig. 5 but for the second episode of the ALADINA campaign, which lasted between 00:00 UTC on 14 May 2018 and 00:00 UTC on 26 May 2018. ALADINA was operated on seven days and data gaps of DMPS data at the Zeppelin Observatory were present from 15:00
UTC on 19 May 2018 until 13:00 UTC on 22 May 2018 as well as between 14:00 UTC on 24 May 2018 and 11:00 UTC on 25 May 2018, mainly within the ALADINA period.

*New version:*

The same parameters as shown in Fig. 5 but for the second episode of the UAS field campaign, when
ALADINA performed research flights on seven different days.

Additional changes made by the authors in the title of Sect. 4:

Concluding remarks  p. 25., l. 498

*was changed to:*

4 Concluding remarks

Additional changes made by the authors in the text of Sect. 4:

*The authors changed a repetition of "it can not be ruled out" that occurred within a short distance of three lines and the text was modified for a clearer statement.*

*The authors made the following changes:*

*old version: p. 26, l. 532-534*

Thus, ABL properties play a significant impact on the vertical and horizontal distribution but **it can not be ruled out** that other sources were simultaneously available during the period, this can not be investigated in detail in case of a restricted investigation area of a few km$^2$.

*new version:*

Thus, ABL properties have a significant influence on the vertical and horizontal distribution but it can not be **excluded** that other sources were present simultaneously during the period, but this can not be investigated in detail in a limited area of a few km$^2$.